



# Rarefied particle motions on hillslopes: 2. Analysis

David Jon Furbish[1], Sarah G. W. Williams[1], Danica L. Roth[2, 3], Tyler H. Doane[1, 4], and Joshua J. Roering[2]

[1]Department of Earth and Environmental Sciences, Vanderbilt University, Nashville, Tennessee, USA
[2]Department of Earth Sciences, University of Oregon, Eugene, Oregon, USA
[3]Current: Department of Geology and Geological Engineering, Colorado School of Mines, Golden, Colorado, USA
[4]Current: Department of Earth and Atmospheric Sciences, Indiana University, Bloomington, Indiana, USA

**Correspondence:** David Furbish (david.j.furbish@vanderbilt.edu)

**Abstract.** We examine a theoretical formulation of the probabilistic physics of rarefied particle motions and deposition on rough hillslope surfaces using measurements of particle travel distances obtained from laboratory and field-based experiments, supplemented with high-speed imaging and audio recordings that highlight effects of particle-surface collisions. The formulation, presented in a companion paper (Furbish et al., 2020a), is based on a description of the kinetic energy balance of a

cohort of particles treated as a rarefied granular gas, and a description of particle deposition that depends on the energy state of the particles. Both laboratory and field-based measurements are consistent with a generalized Pareto distribution of travel distances and predicted variations in behavior associated with the balance between gravitational heating due to conversion of potential to kinetic energy and frictional cooling due to particle-surface collisions. For a given particle size and shape these behaviors vary from a bounded distribution representing rapid thermal collapse with small slopes or large surface roughness,

to an exponential distribution representing approximately isothermal conditions, to a heavy-tailed distribution representing net heating of particles with large slopes. The transition to a heavy-tailed distribution likely involves an increasing conversion of translational to rotational kinetic energy leading to larger travel distances with decreasing effectiveness of collisional friction. This energy conversion is strongly influenced by particle shape, although the analysis points to the need for further clarity concerning how particle size and shape in concert with surface roughness influence the extraction of particle energy and the

likelihood of deposition.

## 1 Introduction

As described in our first companion paper (Furbish et al., 2020a), we are focused on rarefied motions of particles which, once entrained, travel downslope over the land surface. This notably includes the dry ravel of particles down rough hillslopes following disturbances (Roering and Gerber, 2005; Doane, 2018; Doane et al., 2019; Roth et al., 2020) or upon their release from

obstacles (e.g., vegetation) following failure of the obstacles (Lamb et al., 2011, 2013; DiBiase and Lamb, 2013; DiBiase et al., 2017; Doane et al., 2018, 2019), and the motions of rock fall material over the rough surfaces of talus and scree slopes (Gerber and Scheidegger, 1974; Kirkby and Statham, 1975; Statham, 1976; Tesson et al., 2020). By "rarefied motions" we are referring to the situation in which moving particles may frequently interact with the surface, but rarely interact with each other. Thus,



rarefied particle motions are distinct from granular flows. Although this idea is most applicable to processes such as rock fall and the subsequent motions of the rock material over talus or scree slopes, our description of the motions of individual particles nonetheless may be entirely relevant to conditions that are not strictly rarefied, but where during the collective motions of many particles the effects of particle-surface interactions dominate over effects of particle-particle interactions in determining the behavior of the particles — akin to granular shear flows at high Knudsen number (Kumaran, 2005, 2006). We note that laboratory experiments (Kirkby and Statham, 1975; Gabet and Mendoza, 2012) and field-based experiments (DiBiase et al., 2017; Roth et al., 2020) designed to mimic particle motions and travel distances on hillslopes effectively focus on rarefied conditions.

The formulation of rarefied particle motions presented in the first companion paper (Furbish et al., 2020a) is based on a description of the kinetic energy balance of a cohort of particles treated as a rarefied granular gas, and a description of particle deposition that depends on the energy state of the particles. The particle energy balance involves gravitational heating with conversion of potential to kinetic energy, frictional cooling associated with particle-surface collisions, and an apparent heating associated with preferential deposition of low energy particles. Deposition probabilistically occurs with frictional cooling in relation to the distribution of particle energy states as this distribution varies downslope. The Kirkby number $Ki$ — the ratio of gravitational heating to frictional cooling — sets the basic deposition behavior and the form of the probability distribution $f_r(r)$ of particle travel distances $r$. For isothermal conditions where frictional cooling matches gravitational heating plus the apparent heating due to deposition, the distribution $f_r(r)$ is exponential. With non-isothermal conditions and small $Ki$ this distribution is bounded and represents rapid thermal collapse. With increasing $Ki$ the distribution $f_r(r)$ takes the form of a heavy-tailed Pareto distribution. It may possess a finite mean and finite variance with moderate $Ki$, or the mean and variance may be undefined with large $Ki$.

The purpose of this second companion paper is to present an analysis of several data sets concerning particle motions on rough surfaces, as viewed through the lens of the theory presented in the first companion paper (Furbish et al., 2020a). In Section 2 we summarize the context for our work provided by recent probabilistic descriptions of the flux and the Exner equation (Furbish and Haff, 2010; Furbish and Roering, 2013), and then step through essential elements of the mechanical basis of the theory leading to the generalized Pareto distribution of particle travel distances. In Section 3 we compare the formulation with the laboratory measurements of particle travel distances on rough surfaces reported by Gabet and Mendoza (2012) and Kirkby and Statham (1975). We also report laboratory experiments designed to clarify how the size and shape of particles influence their motions and disentrainment based on high-speed imaging. In Section 4 we compare the formulation with the field-based measurements of travel distances reported by DiBiase et al. (2017) and Roth et al. (2020).

Particle travel distances from both the laboratory and field-based experiments are consistent with the generalized Pareto distribution and provide compelling evidence for the full range of predicted behaviors, from rapid thermal collapse to approximately isothermal conditions to net heating of particles. Nonetheless, the analysis points to the need for further clarity concerning how particle size and shape in concert with surface roughness influence the extraction of particle energy and the likelihood of deposition. In the third companion paper (Furbish et al., 2020b) we show that the generalized Pareto distribution in this problem is a maximum entropy distribution (Jaynes, 1957a, 1957b) constrained by a fixed energetic "cost" — the total



cumulative energy extracted by collisional friction per unit kinetic energy available during particle motions. That is, among all possible accessible microstates — the many different ways to arrange a great number of particles into distance states where
each arrangement satisfies the same fixed total energetic cost — the generalized Pareto distribution represents the most probable arrangement. In the fourth companion paper (Furbish et al., 2020c) we step back and examine the philosophical underpinning of the statistical mechanics framework for describing sediment particle motions and transport.

## 2   Key elements of theoretical formulation

### 2.1   Probabilistic description of disentrainment

The problem of describing rarefied particle motions on hillslopes is motivated by the entrainment forms of the flux and the Exner equation. Namely, let $f_r(r; x)$ denote the probability density function of the travel distances $r$ of particles whose motions start at $x$, and let $R_r(r; x)$ denote the associated exceedance probability function. Assuming motions are only in the positive $x$ direction and noting that $x' = x - r$, the flux $q(x)$ may be written as

$$q(x) = \int_{-\infty}^{x} E_s(x') R_r(x - x'; x') \, dx' \,, \tag{1}$$

where $E_s(x)$ denotes the volumetric entrainment rate at position $x$. In turn, letting $\zeta(x, t)$ denote the local land-surface elevation, the entrainment form of the Exner equation is

$$c_b \frac{\partial \zeta(x, t)}{\partial t} = -E_s(x) + \int_{-\infty}^{x} E_s(x') f_r(x - x'; x') \, dx' \,, \tag{2}$$

where $c_b = 1 - \phi_s$ is the volumetric concentration of the surface with porosity $\phi_s$. The central elements of Eq. (1) and Eq. (2) are the probability density function $f_r(r; x)$ and the associated exceedance probability function $R_r(r; x)$. These are related to
the disentrainment rate function defined as

$$P_r(r; x) = \frac{f_r(r; x)}{R_r(r; x)} \,, \tag{3}$$

which, when multiplied by $dr$, is interpreted as the probability that a particle will become disentrained within the small interval $r$ to $r + dr$, given that it "survived" travel to the distance $r$. The disentrainment rate, Eq, (3), connects the descriptions of the flux and its divergence, Eq. (1) and Eq. (2), to the physics of particle motions and disentrainment.

For completeness we note that the formulation above involving continuous functions can be recast into a discrete form that is useful for considering situations in which conditions influencing particle motions, for example the surface slope and roughness texture, change in the downslope direction. Let $k = 1, 2, 3, ...$ denote a set of discrete intervals of length $dr$. Let $p_k$ denote the probability that a particle, having not been disentrained before the $k$th interval, then becomes disentrained within this interval. The probability mass function of particle positions is then

$$f_k(k) = p_k \prod_{i=1}^{k-1} (1 - p_i) \,. \tag{4}$$





The probability $p_k$, like its continuous counterpart $P_r(r; x)$, connects the descriptions of the flux and its divergence to the physics of particle motions and disentrainment. This discrete formulation opens the possibility of describing effects of vary-
ing disentrainment rates in response to changing downslope conditions in a manner intrinsic to particle-based treatments of transport (Tucker and Bradley, 2010), but not readily incorporated in probabilistic descriptions. That is, if surface conditions change in the downslope direction, for example, giving net cooling followed by heating or vice versa, then particles whose travel distances are large enough "see" this change and their behavior concomitantly changes.

As summarized next, the analysis presented in Furbish et al. (2020a) describes the mechanical basis for the disentrainment
rates $P_r(r; x)$ and $p_k$, and the associated probability distributions $f_r(r; x)$ and $f_k(k)$. This involves a consideration of the kinetic energy balance of rarefied particle motions and how this balance determines the deposition of particles in relation to their energy state.

## 2.2   Energy and mass balances

Consider a rough, inclined surface with uniform slope angle $\theta$ (Figure 1). At this juncture we simplify the notation and consider

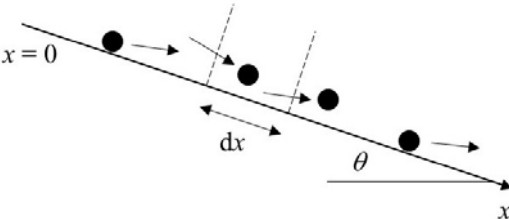

**Figure 1.** Definition diagram of surface inclined at angle $\theta$ and control volume with edge length $\mathrm{d}x$ through which particles move. Figure reproduced from companion paper (Furbish et al., 2020a).

the motions of particles entrained at a single position $x = 0$. Now the particle travel distance $r \to x$ and the probability density function $f_r(r; x) \to f_x(x)$. Consider a control volume with edge length $\mathrm{d}x$ parallel to the mean particle motion. Over a period of time a great number of particles enters the left face of the control volume. Some of these particles move entirely through the volume, exiting its right face, and some come to rest within the control volume. Many, but not necessarily all, of the particles interact with the surface one or more times in moving through the volume or in being deposited within it. We now imagine
collecting this great number of particles and treat them as a cohort, independent of time (Furbish et al., 2020a, Appendix B). That is, let $N(x)$ denote the number of particles that enter the control volume, and let $N(x + \mathrm{d}x)$ denote the number that leaves the volume. The number of particles deposited within the control volume is $\mathrm{d}N = N(x + \mathrm{d}x) - N(x)$. The objective is then to determine the rate of particle deposition, $\mathrm{d}N/\mathrm{d}x$, based on the energy state of the cohort of particles.

In particular, let $E_p = (m/2)u^2$ denote the translational kinetic energy of a particle with mass $m$ and downslope velocity $u$. Letting angle brackets denote an ensemble average of a great number $N$ of particles, then we denote the arithmetic average energy as $E_a = \langle E_p \rangle$ and the harmonic average energy as $E_h = 1/\langle 1/E_p \rangle$. The total energy $E = N E_a$. The formulation





presented in the first companion paper (Furbish et al., 2020a) then leads to four equations with four unknowns involving

energy and mass.

Namely, conservation of the total energy of the particle cohort is given by

$$\frac{\mathrm{d}E(x)}{\mathrm{d}x} = Nmg\sin\theta - Nmg\mu\cos\theta - \frac{1}{\alpha}Nmg\mu\cos\theta. \tag{5}$$

The first term on the right side of Eq. (5) represents gravitational heating with the uniform conversion of potential to kinetic

energy, the second term on the right side represents frictional cooling due to particle-surface collisions, and the last term on the

right side represents a loss of energy associated with particle deposition. The friction factor $\mu$ is

$$\mu = \frac{\langle\beta_x\rangle}{4\tan\phi}, \tag{6}$$

where $\langle\beta_x\rangle$ is the expected proportion of translational energy extracted during a particle-surface collision, and $\phi$ is the expected

reflection angle of particles following collision. The factor $\alpha$ modulates the particle deposition length scale $L_c$, which represents

an $e$-folding distance over which deposition occurs. This length scale is given by

$$L_c = \frac{\alpha E_h}{mg\mu\cos\theta}, \tag{7}$$

and is a function $\alpha = f(Ki)$ of the dimensionless Kirkby number $Ki$ defined by

$$Ki = \frac{mg\sin\theta}{mg\mu\cos\theta} = \frac{S}{\mu}, \tag{8}$$

which represents the ratio of gravitational heating to frictional cooling.

Conservation of particle mass is given by

$$\frac{\mathrm{d}N(x)}{\mathrm{d}x} = -\frac{1}{\alpha E_h}Nmg\mu\cos\theta = -\frac{N}{L_c}, \tag{9}$$

which illustrates that deposition is proportional to frictional cooling depending on the particle energy state $E_h$, modulated by

the factor $\alpha$.

The ensemble averaged energy satisfies the expression,

$$\frac{\mathrm{d}E_a(x)}{\mathrm{d}x} = mg\sin\theta - mg\mu\cos\theta$$

$$+\frac{1}{\alpha}mg\mu\cos\theta\left(\frac{E_a}{E_h}-1\right), \tag{10}$$

where the arithmetic and harmonic averages are related as

$$\frac{E_a}{E_h} = \gamma \geq 1. \tag{11}$$

As with the total energy described by Eq. (5), the first term on the right side of Eq. (10) represents gravitational heating and

the second term on the right side represents frictional cooling. Because the inequality in Eq. (11) must be satisfied, the last

term in Eq. (10) represents an apparent heating due to particle deposition whose effect is to cull lower energy particles, thereby

selecting higher energy particles for continued downslope motion. Brilliantov et al. (2018) describe an analogous behavior of

granular gases due to particle aggregation.





## 5   2.3   Generalized Pareto distribution of particle travel distances

Simultaneous solution of Eq. (5), Eq. (9) and Eq. (10) using Eq. (11) leads to the disentrainment rate function,

$$P_x(x) = \frac{1}{Ax + B}.$$
(12)

In turn the probability density function $f_x(x)$ of travel distances $x$ for particles starting at $x = 0$ is

$$f_x(x) = \frac{B^{1/A}}{(Ax + B)^{1+1/A}},$$
(13)

10   and the exceedance probability function is

$$R_x(x) = \begin{cases} \frac{B^{1/A}}{(Ax+B)^{1/A}} & A \neq 0 \\ e^{-x/B} & A = 0. \end{cases}$$
(14)

The shape and scale parameters $A$ and $B$ are

$$A = \frac{\alpha}{\gamma}\left[\frac{S}{\mu} - 1 + \frac{1}{\alpha}(\gamma - 1)\right] \qquad \text{and}$$
(15)

$$B = \frac{\alpha}{\gamma}\frac{E_{a0}}{mg\mu\cos\theta}.$$
(16)

The mean of the distribution is

$$\mu_x = \frac{B}{1 - A} \qquad A < 1.$$
(17)

With reference to the presentation in the first companion paper (Furbish et al., 2020a) and for the purpose of presenting results below, let $E_{a0}$ denote the initial average particle energy at $x = 0$ and let $N_0$ denote the initial number of particles at
20   $x = 0$. In turn we define a characteristic cooling distance $X = E_{a0}/mg\mu\cos\theta$. For plotting purposes, and to highlight the role of the Kirkby number, we now define the following dimensionless quantities denoted by circumflexes:

$$x = X\hat{x}, \quad N = N_0\hat{N}, \quad E = N_0 E_{a0}\hat{E},$$

$$E_a = E_{a0}\hat{E}_a \quad \text{and} \quad E_h = E_{a0}\hat{E}_h.$$
(18)

With these definitions in place, Eq. (5), Eq. (9), Eq. (10) and Eq. (11) are rewritten as

$$\frac{d\hat{E}(\hat{x})}{d\hat{x}} = \left[Ki - \left(1 + \frac{1}{\alpha}\right)\right]\hat{N},$$
(19)

$$\frac{d\hat{N}(\hat{x})}{d\hat{x}} = -\frac{\hat{N}}{\alpha\hat{E}_h},$$
(20)





$$\frac{\mathrm{d}\hat{E}_a(\hat{x})}{\mathrm{d}\hat{x}} = Ki - 1 + \frac{1}{\alpha}\left(\frac{\hat{E}_a}{\hat{E}_h} - 1\right) \qquad \text{and} \tag{21}$$

$$\frac{\hat{E}_a}{\hat{E}_h} = \gamma \geq 1. \tag{22}$$

The expressions involving energy, Eq. (19) and Eq. (21), reveal that the Kirkby number $Ki$ has a key role in the energy balance

10 and therefore particle deposition. In addition we can define a transitional Kirkby number,

$$Ki_* = 1 - \frac{1}{\alpha}(\gamma - 1). \tag{23}$$

If $Ki < Ki_*$ then net cooling occurs, and if $Ki > Ki_*$ then net heating occurs. The condition $Ki = Ki_*$ implies isothermal conditions such that $\mathrm{d}\hat{E}_a/\mathrm{d}\hat{x} = 0$.

The dimensionless disentrainment rate is

15 $$P_{\hat{x}}(\hat{x}) = \frac{1}{a\hat{x} + b}. \tag{24}$$

The dimensionless probability density function of travel distances is

$$f_{\hat{x}}(\hat{x}) = \frac{b^{1/a}}{(a\hat{x} + b)^{1+1/a}}, \tag{25}$$

and the exceedance probability function is

$$R_{\hat{x}}(\hat{x}) = \begin{cases} \frac{b^{1/a}}{(a\hat{x}+b)^{1/a}} & a \neq 0 \\ e^{-\hat{x}/b} & a = 0. \end{cases} \tag{26}$$

20 The shape and scale parameters $a$ and $b$ are

$$a = A \qquad \text{and} \qquad b = \frac{\alpha}{\gamma}\hat{E}_{a0}. \tag{27}$$

The distribution $f_{\hat{x}}(\hat{x})$ defined by Eq. (25) is a generalized Pareto distribution with position parameter equal to zero (Pickands, 1975; Hosking and Wallis, 1987). To summarize with reference to Figure 2, for $a < 0$ this distribution decays more rapidly than an exponential distribution and is bounded at the position $\hat{x} = b/|a|$. For $a = 0$ it becomes an exponential distribution. For $0 < a < 1/2$ the distribution $f_{\hat{x}}(\hat{x})$ decays more slowly than an exponential distribution, but it possesses a finite mean and a finite variance. For $1/2 \leq a < 1$ the distribution possesses a finite mean but its variance is undefined. For $a \geq 1$ the mean and variance of $f_{\hat{x}}(\hat{x})$ are both undefined, even though this distribution properly integrates to unity. For $a > 0$ the tail of $f_{\hat{x}}(\hat{x})$ decays as a power function, namely, $f_{\hat{x}}(\hat{x}) \sim \hat{x}^{-(1+1/a)}$. The exceedance probability decays as $R_{\hat{x}}(\hat{x}) \sim \hat{x}^{-1/a}$. These results

are summarized in Table 1. If the shape and scale parameters $a$ and $b$ are redefined as $a_L = 1/a$ and $b_L = b/a = ba_L$, then Eq. (25) becomes

$$f_{\hat{x}}(\hat{x}) = \frac{a_L b_L^{a_L}}{(\hat{x} + b_L)^{1+a_L}} \qquad a_L, b_L > 0, \tag{28}$$





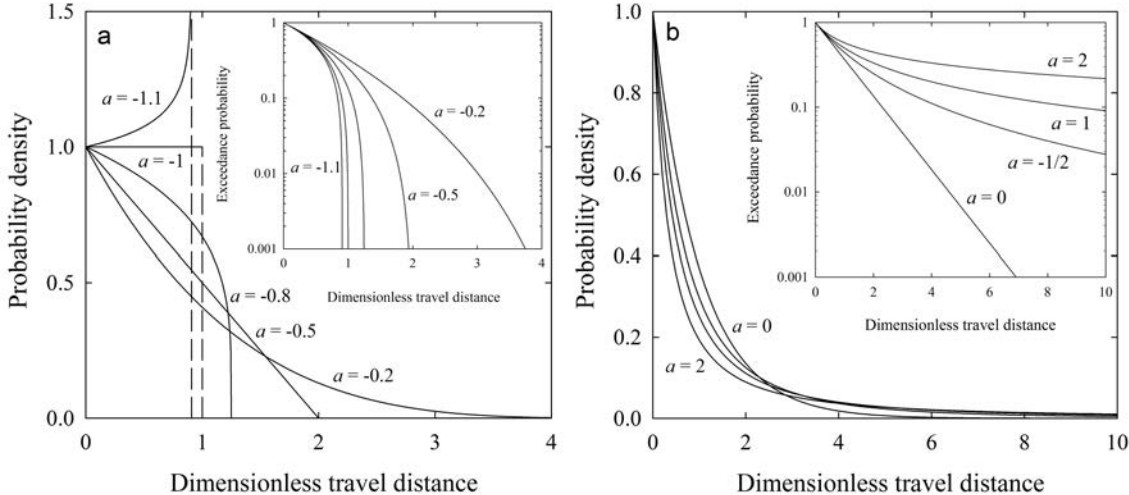

**Figure 2.** Plot of dimensionless probability density $f_{\hat{x}}(\hat{x})$ versus dimensionless travel distance $\hat{x}$ for scale parameter $b = 1$ and different values of the shape parameter $a$ for (a) $a < 0$ and (b) $a \geq 0$ with associated exceedance probability plots (insets). Figure reproduced from companion paper (Furbish et al., 2020a). Compare with Figure 1 in Hosking and Wallis (1987).

**Table 1.** Behavior of the generalized Pareto distribution associated with the shape parameter $a$ and Kirkby number $Ki$ as illustrated in Figure 2.

| Behavior | Range of $a$ | Range of $Ki$ | Mean $\mu_{\hat{x}}$ | Variance $\sigma_{\hat{x}}^2$ |
|---|---|---|---|---|
| Bounded[1], increasing with $\hat{x}$ | $a < -1$ | $Ki < 1 - (2\gamma - 1)/\alpha$ | $b/(1-a)$ | $b^2/(1-a)^2(1-2a)$ |
| Uniform | $a = -1$ | $Ki = 1 - (2\gamma - 1)/\alpha$ | $b/2$ | $b^2/12$ |
| Bounded[1,2], decreasing with $\hat{x}$ | $-1 < a < 0$ | $Ki < Ki_* = 1 - (\gamma - 1)/\alpha$ | $b/(1-a)$ | $b^2/(1-a)^2(1-2a)$ |
| Exponential | $a = 0$ | $Ki = Ki_* = 1 - (\gamma - 1)/\alpha$ | $b$ | $b^2$ |
| Finite mean and variance | $0 < a < 1/2$ | $Ki_* < Ki < Ki_* + \gamma/2\alpha$ | $b/(1-a)$ | $b^2/(1-a)^2(1-2a)$ |
| Finite mean, undefined variance | $1/2 \leq a < 1$ | $Ki_* + \gamma/2\alpha \leq Ki < 1 + 1/\alpha$ | $b/(1-a)$ | — |
| Undefined mean and variance | $a \geq 1$ | $Ki \geq 1 + 1/\alpha$ | — | — |

[1] Truncation occurs at dimensionless distance $\hat{x} = b/|a|$.
[2] Triangular with $a = -1/2$.

which is a Lomax distribution with mean

$$\mu_{\hat{x}} = \frac{b_L}{a_L - 1} \qquad a_L > 1. \tag{29}$$

With $a < 0$ the bounded form of $f_{\hat{x}}(\hat{x})$ (Figure 2) represents rapid thermal collapse with net frictional cooling. For $a = 0$ the exponential form of this distribution represents isothermal conditions where frictional cooling is matched by gravitational





heating and the apparent heating due to deposition. For $a > 0$ the heavy-tailed form of $f_{\hat{x}}(\hat{x})$ represents net gravitational heating.

For reference to data fitting presented below, a binomial expansion of Eq. (13) gives

$$f_x(x) = \frac{1}{B}\left[1 - \frac{A}{B}\left(1 + \frac{1}{A}\right)x - ...\right]. \tag{30}$$

The expansion of an exponential distribution with mean $\mu_x$ is

$$f_x(x) = \frac{1}{\mu_x}\left(1 - \frac{x}{\mu_x} + ...\right). \tag{31}$$

These two expansions indicate that unless the travel distance data span a significant proportion of the $x$ domain, then at lowest order the fit of the generalized Pareto distribution looks like an exponential distribution with mean $B$. This result also is obtained from the disentrainment rate function, Eq. (12), in which for small $x$ this function becomes $P_x \approx 1/B \approx 1/\mu_x$. Moreover, if the travel distance data sample over a majority of the probability contained in the distribution, and if the tail of the distribution is not "too" heavy, then $B$ is an approximation of the mean (where $A < 1$ such that the mean exists).

Also note that in fitting the data to the generalized Pareto distribution, Eq. (13), we use the dimensional form of the exceedance probability, Eq. (14). Specifically, we estimate values of the exceedance probability as $R_x(x) = 1 - r_x/(N+1)$, where $r_x$ is the ascending rank of each datum. We then visually fit Eq. (14) to these estimated values to obtain values of the parameters $A$ and $B$. This involves iteratively examining the data and theoretical lines in arithmetic, semi-log and log-log plots, noting that semi-log plots generally highlight deviations in the tails whereas log-log plots highlight deviations near the origin. We mostly pay attention to the main body of data in the semi-log plots, avoiding over-fitting of the outer part of the tails given the inherent variability of estimates of the exceedance probability associated with the tails, notably with small sample sizes (Appendix A). We also examine the data using quartile-quartile (QQ) plots to ensure that these are consistent with the generalized Pareto distribution. Here we note that our objective is to demonstrate that the data are consistent with the several forms of the generalized Pareto distribution, where in semi-log space the exceedance plots either: 1) have negative concavity (representing rapid thermal collapse); 2) are approximately straight (representing isothermal conditions); or 3) have positive concavity (representing net heating). We are aimed at reasonable estimates of the shape and scale parameters in order to achieve this objective, but we do not need refined estimates of these quantities. For reasons that are fully explained in Appendix A, we therefore use estimates of $A$ and $B$ obtained from visual fitting, avoiding known limitations of quantitative estimates (e.g., maximum likelihood estimates) associated with small sample sizes, particularly in the presence of censorship. Assuming a value of $\gamma$ (see description below), we then have two equations, Eq. (15) and Eq. (16), with two unknowns, $\mu$ and $\alpha$. Thus, the fitting of $A$ and $B$ provides estimates of $\mu$ and $\alpha$ for subsequent consideration. In particular, we first estimate $\mu$ as

$$\mu = S - \frac{E_{a0}(A - 1 + 1/\gamma)}{Bmg\cos\theta}, \tag{32}$$

and then obtain $\alpha$ as

$$\alpha = \frac{B\gamma mg\mu\cos\theta}{E_{a0}}. \tag{33}$$

In turn the Kirkby number $Ki$ is calculated from Eq. (8).





## 2.4 Elements of collisional friction

The quantities $\mu$ and $\alpha$ defined in relation to Eq. (6) and Eq. (7) merit further description. We start by noting that the formulation summarized in the preceding section is based on the assumption that a change in translational kinetic energy $\Delta E_p$ associated with a particle-surface collision can be expressed as $\Delta E_p = -\beta_x E_p$ so that $\beta_x = -\Delta E_p/E_p$ is the proportion of the energy extracted during the collision. Both $\Delta E_p$ and $\beta_x$ are random variables. As described in Appendix E of the first companion paper (Furbish et al., 2020a), in general we may write the energy balance of a particle as

$$\Delta E_p = -\Delta E_r - f_c - f_y. \tag{34}$$

Here, a positive change in rotational energy $\Delta E_r$ is seen as an extraction of translational energy. This loss of translational energy with the onset of rotation may be relatively large if a collision involves stick following initial sliding due to a large normal impulse, and such a loss also may occur due to the imposed torque of friction during a collision that does not necessarily involve stick. The term $f_c$ in Eq. (34) represents losses associated with particle and surface deformation as well as work performed against friction during collision impulses (thence converted to heat and sound). But this term also includes losses associated with deformation of the surface at a scale larger than that of an idealized particle-surface impulse contact, namely, due to momentum exchanges associated with the sputtering of loose surface particles during collision. (The videos published as supplementary material to DiBiase et al. (2017) nicely illustrate this sputtering as well as the onset of rotational motion.) The term $f_y$ in Eq. (34) represents the energy loss associated with glancing collisions that produce transverse translational motions and rotation oriented differently than any incident rotation. In some cases the change in energy $\Delta E_p$ can be expressed directly in terms of the energy state $E_p$ (Appendix E in Furbish et al., 2020a). However, the complexity of particle-surface collisions on natural hillslopes precludes explicitly demonstrating such a relation for all possible scenarios. Nonetheless, it is entirely defensible to assume that energy losses can be related to the energy state $E_p$ if the elements involved are formally viewed as random variables. Then, the simple relation $\Delta E_p = -\beta_x E_p$ is to be viewed as an hypothesis to be tested against data.

This hypothesis formally enters the formulation via Eq. (6). Namely, from this relation we may write $\mu \sim \langle \beta_x \rangle$, highlighting that $\mu$ is associated with the cooling rate. In turn, particle collision mechanics (Appendix E in Furbish et al., 2020a) suggest, for example, that $\mu \sim \langle \beta_x \rangle \sim M(\theta)$, where $M(\theta)$ involves the coefficients associated with tangential and normal impulses contributing to energy losses during collisions, and depends on the slope angle $\theta$ in that the expected surface normal impact velocity varies with this angle. (In an idealized particle-surface collision these coefficients include the normal coefficient of restitution and a coefficient describing the ratio of tangential to normal impulses during the collision (e.g., Brach, 1991; Brach and Dunn, 1992, 1995)). Moreover, $M(\theta)$ is independent of particle size. These points are examined below in relation to experimental data.

In turn, Furbish et al. (2020a) suggest that the quantity $\alpha$ is related to the Kirkby number $Ki$ as

$$\alpha = \frac{\alpha_0}{1 - \mu_1 Ki}, \tag{35}$$





where $\alpha_0$ denotes the value of $\alpha$ associated with a flat surface ($Ki = 0$) and $\mu_1$ is a factor of order unity. Substitution into Eq. (7) gives

$$L_c = \frac{\alpha_0 E_a}{mg\mu\cos\theta - mg\mu_1\sin\theta}\,.$$
(36)

Viewed in this manner, $\alpha$ represents a direct effect of heating described by $mg\mu_1\sin\theta$, namely, to decrease the likelihood of deposition by decreasing the proportion of particles that cool to sufficiently low energies for deposition to occur — which translates to suppressing the disentrainment rate and increasing the length scale of deposition $L_c$ relative to the cooling length scale $l_c = E_h/mg\mu\cos\theta$. In this view, $\mu$ goes with the cooling rate (not the deposition rate). But we also may write Eq. (36)

as

$$L_c = \frac{\alpha_0 E_h}{mg\cos\theta\mu(1 - \mu_1 Ki)}$$
(37)

Viewed in this manner, we may define an apparent friction factor as $\mu_0 = \mu(1 - \mu_1 Ki)$ associated with deposition. Here again, $\mu$ is associated with the cooling rate but is then modulated by heating. We suggest below that for the same particle size, $\alpha$ increases with increasing $Ki$, very likely due to a combination of increased heating and increased partitioning of translational
energy to rotational motion — both decreasing the likelihood of stopping and not represented in just the factor $\mu$. We also suggest that for the same slope and surface roughness, $\alpha$ increases with increasing particle size, decreasing the likelihood of frictional loss with increasing rotational energy.

## 3 Laboratory measurements

### 3.1 Gabet and Mendoza (2012)

#### 3.1.1 Experiments

The experiments reported by Gabet and Mendoza (2012) involved launching spherical, sub-angular one-cm particles with initial velocity $u_0 = 0.7$ m s$^{-1}$ onto an inclined surface with fixed roughness elements embedded within concrete (see Figure 2 therein). The experiments stepped through slope angles of $\theta = 0$ degrees to $\theta = 30$ degrees in increments of three degrees. The travel distances of 100 particles were measured for each slope angle. All 100 particles stopped within the 3 m flume for angles 0 to 15 degrees. For angles 18, 21, 24 and 27 degrees, 92, 58, 26 and 4 particles stopped, respectively. No particles stopped on the 30 degree slope.

Because the same particle is launched each time with the same initial velocity $u_0$, the initial arithmetic and harmonic averages of the particle energy are the same, that is, $E_{a0}/E_{h0} = \gamma_0 \approx 1$. Over some unknown distance the particle motions experience
randomization via collisions such that $E_a/E_h = \gamma > 1$. With reference to Eq. (9) where $E_h = E_a/\gamma$, this means that because $\gamma$ is initially one and then increases, the expected disentrainment rate likely increases initially over a short distance. Indeed, in preliminary plots (not shown) of the exceedance probability $R_x(x)$ versus $x$, an inflection occurs in some of the data close to $x = 0$, which we suspect represents a delay in the onset of deposition of the lowest energy particles. In fitting the data to





the exceedance probability $R_x(x)$ we assume that $E_a/E_h = \gamma > 1$ (and fixed) over the entirety of the travel distances. For
this reason we truncate distances shorter than the inflection position then recalculate the travel distances and the exceedance
probabilities, while avoiding large truncation given the limited data size. We cannot know for certain the appropriate truncation
position, so this point should be kept in mind. Note, however, that the formulation does not care where motions start, so this
truncation is just a resetting of the staring position ($x = 0$) with fewer data, assuming $\gamma$ remains approximately fixed beyond the
adjusted starting position. These points are examine further in Appendix B with reference to experiments described in Section
15   3.3.

We choose $\gamma = 1.5$ in this and subsequent analyses of travel distance data. Note first that this choice has no influence on the
estimates of the parameters $A$ and $B$. However, it does influence the estimated values of $\mu$ and $\alpha$ via Eq. (32) and Eq. (33).
The assumption that $\gamma$ is fixed may be incorrect. However, to rigorously constrain $\gamma$ would require solving the Fokker-Planck
equation describing the evolving number density $n_{E_p}(E_p, x)$ of particle energy states $E_p$ as described in Section 3.3.2 of the
first companion paper (Furbish et al., 2020a); this is beyond the scope of our objective of demonstrating the basic behavior
of the particle energy balance. The effect of fixed $\gamma = 1.5$ on estimates of $\mu$ and $\alpha$ is systematic throughout the analyses, but
whether this choice is correct remains an open question.

### 3.1.2   Results

The data are reasonably well fit by the theoretical curves, where we plot the data twice (Figure 3 and Figure 4)  in order to

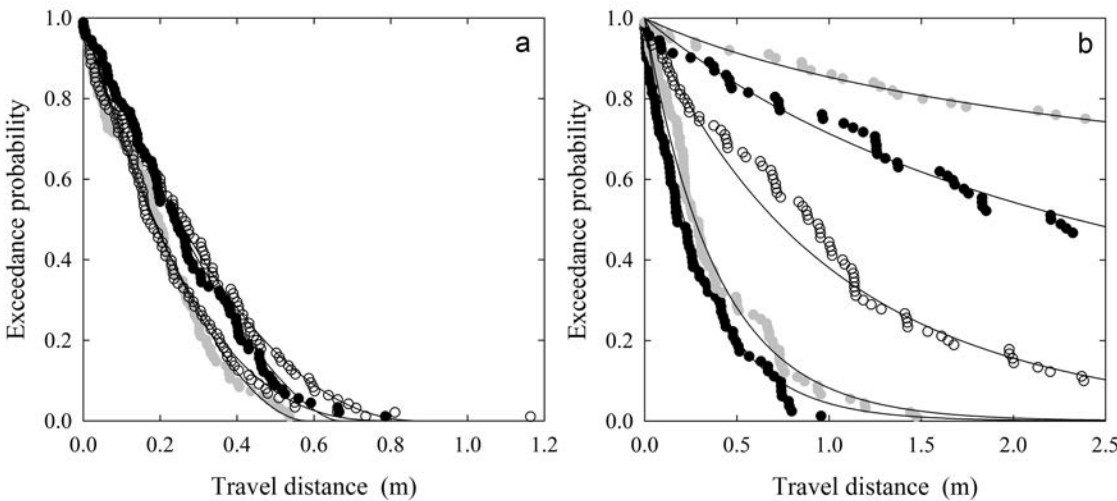

**Figure 3.** Plot of exceedance probability $R_x(x)$ versus travel distance $x$ for experiments described by Gabet and Mendoza (2012) where (a)
$A < 0$ and (b) $A \geq 1$.

highlight several points. Estimated parametric values are provided in Table 2, where estimates of the variability in $\mu$, $\alpha$, $Ki$ and
$Ki_*$ are based on a Monte Carlo analysis as described in Appendix C.



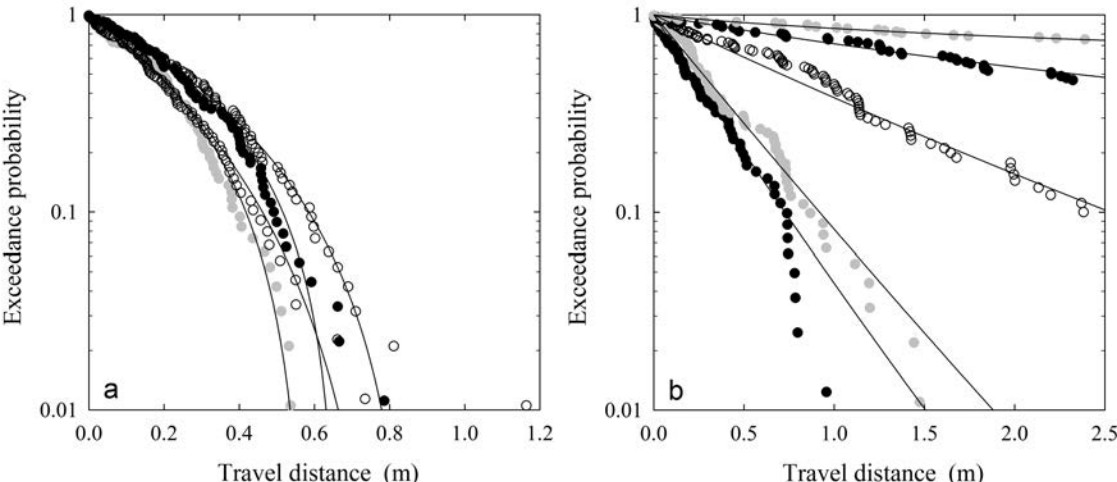

**Figure 4.** Plot of logarithm of exceedance probability $R_x(x)$ versus travel distance $x$ for experiments described by Gabet and Mendoza (2012) where (a) $A < 0$ and (b) $A \geq 0$, together with fitted distributions (lines).

**Table 2.** Fitted and estimated values of the parameters for the data shown in Figure 3 and Figure 4 with coefficients of variation in parentheses.

| Slope (deg) | $A$ | $B$ (m) | $Ki$ | $Ki_*$ | $\mu$ | $\alpha$ | $\mu_x^a$ (m) | $\mu_x^b$ (m) |
|---|---|---|---|---|---|---|---|---|
| 0 | -0.48 | 0.42 | 0.00 (—)[c] | 0.59 (0.042) | 0.05 (0.12) | 1.22 (0.061) | 0.28 | 0.29 |
| 3 | -0.55 | 0.32 | 0.43 (0.070) | 0.78 (0.017) | 0.12 (0.073) | 2.33 (0.056) | 0.21 | 0.21 |
| 6 | -0.36 | 0.30 | 0.64 (0.044) | 0.83 (0.016) | 0.16 (0.046) | 2.89 (0.073) | 0.22 | 0.22 |
| 9 | -0.65 | 0.43 | 0.73 (0.036) | 0.91 (0.0082) | 0.22 (0.035) | 5.53 (0.080) | 0.26 | 0.27 |
| 12 | 0.03 | 0.31 | 0.89 (0.012) | 0.88 (0.014) | 0.23 (0.013) | 4.26 (0.10) | 0.31 | 0.29 |
| 15 | 0.02 | 0.39 | 0.93 (0.0078) | 0.92 (0.0084) | 0.29 (0.0079) | 6.53 (0.098) | 0.40 | 0.40 |
| 18 | 0.09 | 0.99 | 0.98 (—) | 0.97 (—) | 0.33 (—) | 18.7 (0.10) | 1.09 | 0.85 |
| 21 | 0.70 | 2.6 | 1.01 (—) | 0.99 (—) | 0.38 (—) | 56.1 (0.10) | 8.77 | 1.06 |
| 24 | 3.6 | 4.7 | 1.04 (0.0069) | 1.00 (—) | 0.43 (0.0067) | 110 (0.11) | — | 0.98 |

[a] Estimated from parameters $A$ and $B$.

[b] Estimated from data, recognizing that these do not incorporate distances of particles that moved beyond measurement distance of 3 m.

[c] Notation (—) means undefined or coefficient of variation is less than 0.01.

For data involving an estimated shape parameter $A < 0$ (Figure 3a, Figure 4a), the relatively rapid decrease in the exceedance probability $R_x(x)$ with little indication of an asymptotic approach to $R_x(x) = 0$ strongly suggests that the data represent bounded forms of the distribution $f_x(x)$ (Figure 3a). Nonetheless, one might on empirical grounds reasonably fit the data to, say, an exponential distribution (although quartile-quartile plots, not shown, would advise against this). However, the negative concavity of the fits are unambiguous in Figure 4a, strongly reinforcing the point that the data represent bounded forms of the





distribution. This result is consistent with the idea of rapid thermal collapse for data involving $\theta \leq 9$ degrees and $Ki \leq 0.73$. Note that $Ki < Ki_*$ in all cases. For unknown reasons (and not attributable to truncation) the average particle motion on the flat surface is larger than the next three slope angles (3, 6 and 9 degrees). This may simply reflect the stochasticity of the disentrainment process at these small angles. Also note that several data sets share "kinks" in their estimated exceedance probabilities at similar distances, for example, around 0.8 m and 1.3 m. This could be due to chance, or it may reflect a persistent effect of the structure of the surface roughness, specifically the occurrence of relatively large roughness elements. Roth et al.

(2020) note this behavior in their field-based measurements of particle travel distances on vegetated hillslopes (Section 4.2), where vegetation acts as roughness elements.

For data involving an estimated shape factor $A \geq 0$ (Figure 3b, Figure 4b), the first two sets (12 and 15 degrees) are approximately exponential with $Ki \approx Ki_*$. This is reflected in the close correspondence between the values of the scale parameter $B$ and the estimated average travel distances $\mu_x$ (Table 2). The data show a clear asymptotic appearance of the exceedance

probability $R_x(x)$ (Figure 3b) with essentially straight line fits in Figure 4b. This result is consistent with the idea that these two experiments involved approximately isothermal conditions. For larger shape factor $A$, the fitted lines decrease in an exponential-like manner in Figure 3b and they appear as essentially straight lines over the domain of measured travel distances in Figure 4b. The fits therefore cannot be used to distinguish between an exponential distribution and a generalized Pareto distribution. Note, however, that regardless of the distribution, whereas the data for 18 degrees span about 90% of the probability of the

distribution, the data for 21 degrees sample only about 50% of the probability contained in the distribution, and the data for 24 degrees sample only about 15% of the probability. This directly points to the difficulty of working with tail-censored data, especially if the underlying distribution is heavy-tailed (Appendix A). That is, at large $Ki$ the experimental flume is sampling only a fraction of the distribution, representing just the shortest travel distances associated with the specific surface-roughness conditions. Nonetheless, the mechanical basis of the distribution combined with its consistency with data for $A < 0$ reinforces

the merit of the hypothesis of a heavy-tailed behavior for $A > 0$.

Further note that for all cases less than 24 degrees the estimated values of $A$ and $B$ suggest that the distributions have finite moments. These moments are undefined ($A > 1$) for the case of 24 degrees. Also recall that only four particles stopped on the flume at a slope angle of 27 degrees, and no particles stopped at an angle of 30 degrees. These points are consistent with the idea that gravitational heating is systematically increasing relative to frictional cooling with increasing Kirkby number. Using

the largest estimated value of $\mu$ (Table 2), the values of the Kirkby number would be $Ki = 1.2$ and $Ki = 1.3$ for the slope angles of 27 and 30 degrees, respectively.

Here is an interesting sidebar. Following Samson et al. (1999) we plot the reciprocal, $1/\mu_x$, of the average travel distance $\mu_x$ versus slope angle $\theta$ (Figure 5). For spheres rolling bumpety-bump down an inclined surface roughened with a quasi-random monolayer of small particles, Samson et al. (1999) show a linear decline in $1/\mu_x$ with increasing slope angle (see their Figure

3) associated with trapping (deposition) related to collisional friction. This reciprocal then smoothly transitions to values close to zero with further increase in slope angle representing continuing motions without significant trapping. The plot in Figure 5 also reinforces the idea that if one assumes an exponential distribution to calculate average travel distances, the values will





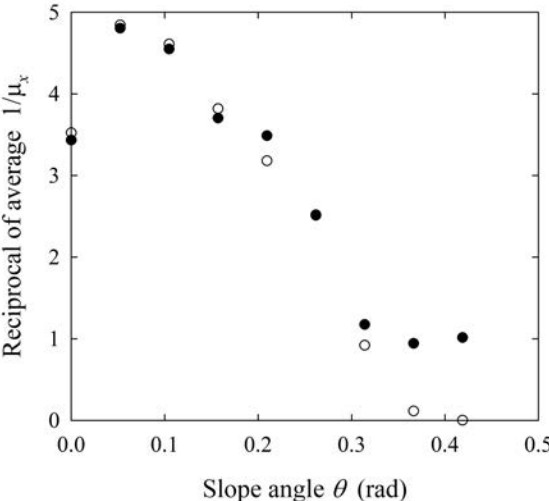

**Figure 5.** Plot of reciprocal of average travel distance $1/\mu_x$ versus slope angle $\theta$ calculated from Eq. (17) using estimated values of $A$ and $B$ (open circles) and calculated directly from data (black circles).

be similar to those associated with the generalized Pareto distribution for small to moderate Kirkby numbers, but then deviate significantly with increasing heaviness of the distribution tail.

## 3.2 Kirkby and Statham (1975)

### 3.2.1 Experiments

The experiments reported by Kirkby and Statham (1975) involved dropping particles onto an inclined surface with fixed roughness composed of particles of different sizes, thus giving different ratios of particle to roughness size. For different slopes, the particles were dropped from different initial heights $h$ at the crest of the surface, and travel distances were measured. Here we focus on average travel distances reported in their Figure 1 involving a slope angle of 35 degrees and three dropped particle sizes with minor-to-intermediate axis ratios of 0.40, 0.53 and 0.77. Kirkby and Statham (1975) report that the distributions of travel distances are exponential, but do not present the travel distance data.

For $A$ and $B$ defined by Eq. (15) and Eq. (16) we may write the average travel distance as

$$\mu_x = \frac{B^*}{1 - A}\epsilon^2 h, \tag{38}$$

where $B^* = (\alpha/\gamma)\sin^2\theta/\mu\cos\theta$ or $B = B^*\epsilon^2 h$. The form of Eq. (38) requires a straight line fit between the average distance $\mu_x$ and the drop height $h$, but it does not provide a unique fit. We must ensure that the estimated shape parameter $A < 1$ to yield a finite average; otherwise the comparison is not meaningful. Based on the results described in the previous section we assume that the Kirkby number is close to unity, and for the given slope angle we choose a friction factor of $\mu = 0.7$. Whereas the coefficient of restitution $\epsilon$ may be relatively large for natural rock material, this coefficient typically decreases significantly




on average for irregular particles due to the high probability that collisions are not collinear with particle centers of mass (see next section). For illustration we choose $\epsilon = 0.5$. As before we fix $\gamma$ and then vary $\alpha$ to estimate $A$ and $B^*$.

### 3.2.2 Results

The data are well fit by Eq. (38) (Figure 6). Estimated parametric values are provided in Table 3.

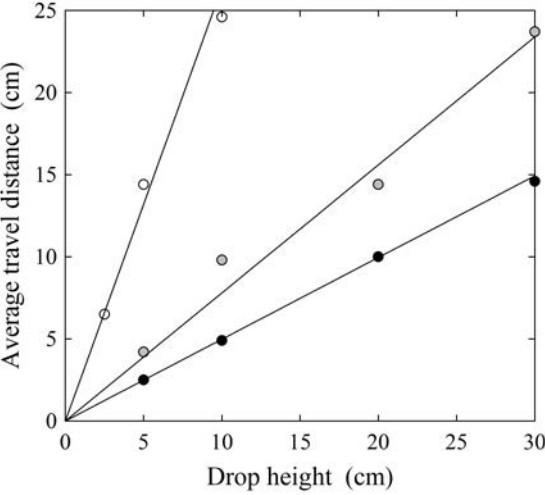

**Figure 6.** Plot of average travel distance $\mu_x$ versus drop height $h$ based on data in Figure 1 of Kirkby and Statham (1975) for three different particle sizes. Note that we are assuming the largest size is 21.5 mm rather than the reported value of 0.215 mm.

**Table 3.** Fitted values of the parameters for the data shown in Figure 6.

| Particle size (cm) | $A$ | $B^*$ | $Ki$ | $\mu$ | $\alpha$ |
|---|---|---|---|---|---|
| 1.26 | 0.33 | 1.3 | 1.0 | 0.7 | 3.5 |
| 1.34 | 0.33 | 2.1 | 1.0 | 0.7 | 5.5 |
| 2.15 | 0.31 | 7.2 | 1.0 | 0.7 | 19 |

We emphasize that other choices of the quantities $\epsilon$, $\mu$ and $\alpha$ would provide equally good fits, given that Eq. (38) does not provide a unique solution for $A$ and $B^*$. With this in mind, the estimated values of $A$ suggest that the data are consistent with a heavy-tailed form of the generalized Pareto distribution with finite mean and finite variance. The dropped particles experience different rates of frictional cooling, manifest in the increasing value of $\alpha$ with increasing particle size (Table 3); and the data are consistent with the idea that the average travel distance is directly proportional to the initial energy state determined by the
drop height $h$.





### 3.3 Vanderbilt data

#### 3.3.1 Experiments

We conducted two sets of experiments. The first set was aimed at demonstrating the basis for treating the proportion of energy extraction, $\beta_x$, as a random variable. To do this we focused on the analogous quantity $\beta_z$, which is the proportion of energy extraction associated with particle collision on a horizontal surface following vertical free fall. This allowed us to show, and calculate, the partitioning of translational energy into deformational friction and rotational energy during collisions. We used particles of varying size and angularity. The experiments involved dropping the particles onto a smooth rigid surface of slate, and onto a rough surface. The rough surface, made of concrete, had a granular roughness smaller than the particles (Figure 7). We recorded particle motions using a Lightning RDT monochrome camera (DRS Technologies) operating at 800 frames per

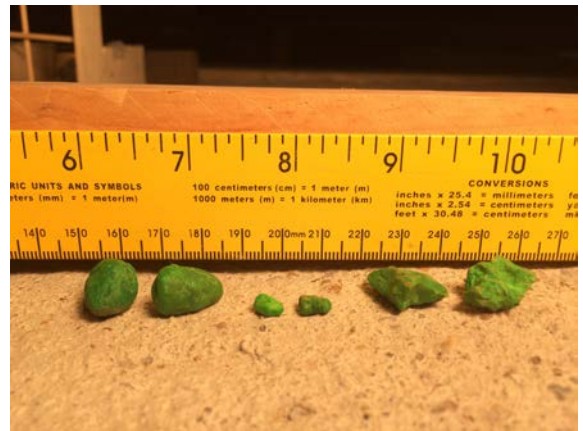

**Figure 7.** Image of rounded (left), small (center) and angular (right) test particles on the concrete surface with granular roughness texture.

second. The camera was mounted on a tripod, and oriented parallel to the horizontal surface. The image resolution was 1,280 $\times$ 640 pixels.

In the second set of experiments we launched particles of varying size and angularity onto the rough surface, then measured their travel distances for several slope angles. The slopes were $S = 0, 0.09, 0.15, 0.18, 0.25$ and $0.28$. The launching
10  device consisted of a pendulum catapult (Figure 8) configured so that particles were delivered to the rough surface with negligible rotational motion and with a prescribed surface-parallel velocity. We used two sizes ($D \approx 1$ cm and $D \approx 0.5$ cm) of irregular, rounded particles and one size ($D \approx 1$ cm) of angular particles. We recorded motions of particles launched from the catapult onto the rough surface with high-speed imaging at a resolution of $640 \times 640$ pixels. In addition we made audio recordings of particle-surface interactions during their downslope motions. Audio recordings were made and processed using
15  the GarageBand application on a 6th generation Apple iPad.





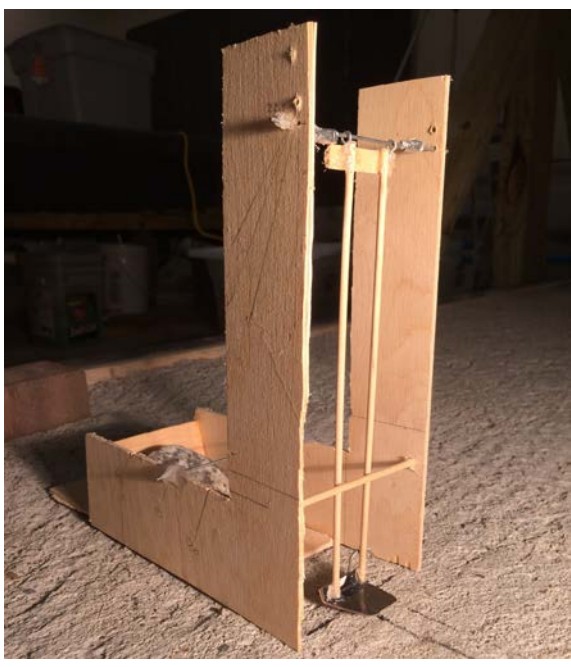

**Figure 8.** Image of pendulum catapult. A particle is placed on the low friction (glossy cardboard) cradle at the base of the pendulum arms ($\sim 20$ cm); using a wand the arms and particle are pushed back to a preset position as one would a toddler on a playground swing (albeit not with a wand), then released; the arms are arrested by the front bar, whence the momentum of the particle launches it onto the surface. The cradle is about 2 mm above the rough surface. A stone is placed in the base of the catapult for stability.

### 3.3.2 Results

As a point of reference, the vertical rebounds of ordinary spherical glass marbles following their impacts on the hard slate reveal no surprises. These collinear collisions give a normal coefficient of restitution of $\epsilon = 0.81 \pm 0.017$ yielding $\beta_z = (1 - \epsilon^2) = 0.34 \pm 0.028$ ($m = 0.014$ Kg, $N = 5$), and $\epsilon = 0.81 \pm 0.018$ yielding $\beta_z = 0.35 \pm 0.030$ ($m = 0.0033$ Kg, $N = 5$). The variation in $\epsilon$ is likely attributable to small differences in the marble-surface deformation mechanics during collision. Rebounds from the rigid granular surface give $\epsilon = 0.26 \pm 0.59$ yielding $\beta_z = 0.93 \pm 0.031$ ($m = 0.014$ Kg, $N = 10$), and $\epsilon = 0.41 \pm 0.038$ yielding $\beta_z = 0.83 \pm 0.032$ ($m = 0.030$ Kg, $N = 10$). Although the collisions are collinear, the granular texture of the surface leads to some variation in the reflection angles. The smaller values of $\epsilon$ relative to the slate surface indicate that, although the concrete surface is rigid, its granular texture gives more dissipative collisions, likely involving deformation of micro-asperities. This

effect evidently is more pronounced with the larger marbles.

In contrast, the rebounds of natural particles from the hard slate reveal how noncollinear collisions strongly influence the rebound angle and normal rebound height. If $\epsilon_z$ denotes the effective normal coefficient of restitution, then $\beta_z = 1 - \epsilon_z^2$ is the proportion of the normal (vertical) component of kinetic energy extracted, analogous to $\beta_x$. That is, $\beta_z = -\Delta E_p / E_p$. This proportion involves a mechanical loss due to particle and surface deformation during the collision, with conversion of energy to





heat and sound. But a significant part can go into transverse components of translational energy and rotational energy. Thus, this is a simple demonstration of the idea that $\beta_z$ (and $\beta_x$) must be treated as a random variable rather than a fixed (deterministic) quantity as in the case of the normal coefficient of restitution $\epsilon$ associated with spherical particles in a granular gas.

To illustrate this we write a normalized form of $\beta_z$ as $\beta_z^* = (\beta_z - \beta_z^{\min})/(1 - \beta_z^{\min})$, where $\beta_z^{\min} = 1 - \epsilon^2$ is the minimum value associated with a collinear collision. Although we cannot offer a theoretical basis for the distribution of $\beta_z^*$, we nonethe-

less on empirical grounds fit $\beta_z^*$ to the beta distribution because of its versatile bounded form over $[0, 1]$, then transform the fitted distribution back to the original values of $\beta_z$. For comparison we fit a Gaussian distribution to the values $\beta_z$ measured for the spheres.

For the hard slate surface, spheres show a small variance about the mean value. Probability is then redistributed toward $\beta_z = 1$ with increasing particle angularity (Figure 9). For the rough surface, spheres show a larger variance, and there is a

stronger redistribution of probability toward $\beta_z = 1$ with increasing angularity (Figure 10). We cannot directly map this result to an interpretation of $\beta_x$ because of differences in the geometrical conditions of collisions. Nonetheless, as described below, the effect of angularity appears in measurements of travel distances as an increasing likelihood of disentrainment with increasing angularity.

For the rough experimental surface, $\beta_z$ is on average larger than for the hard slate surface. The particle-surface impact is

unlike that of an idealized rigid surface, and more like that of a quasi-rigid (deformable) granular material. Nonetheless, despite the small effective coefficient of restitution, noncollinear collisions yield significant conversion of energy to transverse motions and rotation. The particles do not simply "die" on impact.

For each particle-surface combination, the largest rebound heights provide an estimate of the (ordinary) normal coefficient of restitution $\epsilon$. These heights are associated with approximately collinear collisions as confirmed by the high-speed imaging. We

can therefore estimate the partitioning of energy between the frictional loss $f_c$ (deformation, heat and sound) and reflectional transverse motion and rotation $E_c$ (Appendix D). Results indicate that on average less than half of the initial energy is dissipated by friction on the hard surface, and slightly less goes to transverse motion and rotation (Table 4). About 20-30% of the initial

**Table 4.** Average energy partitioning as a proportion of initial energy $E_{p0} = mgh$ for estimated coefficient of restitution $\epsilon$.

| Surface | Particle shape | $\epsilon$ | $f_c$ | $E_c$ | Recovered |
|---------|---------------|------------|-------|-------|-----------|
| hard slate | rounded | 0.80 | 0.36 | 0.34 | 0.30 |
|  | angular | 0.75 | 0.44 | 0.35 | 0.21 |
| rough concrete | rounded | 0.42 | 0.82 | 0.13 | 0.05 |
|  | angular | 0.37 | 0.86 | 0.10 | 0.04 |

energy is recovered in vertical motion. In contrast, on average much of the initial energy is dissipated by friction on the rough surface, and only about 10% or so goes to transverse motion and rotation. About 5% of the initial energy is recovered in vertical motion. These results are consistent with those reported by Williams and Furbish (2020) involving a larger data set.

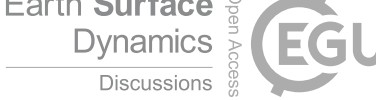

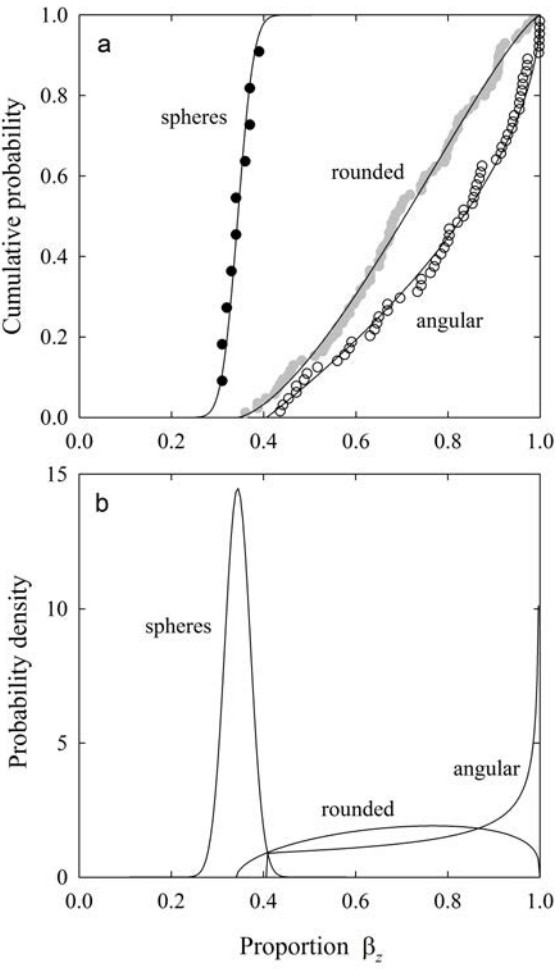

**Figure 9.** Plot of (a) cumulative distributions of $\beta_z$ for glass spheres fit to a Gaussian distribution and rounded and angular particles fit to a beta distribution, and (b) associated probability density functions of fitted distributions. Collisions are on hard slate.

High-speed imaging of particle motions launched by the catapult onto the rough surface provides estimates of initial surface-parallel particle velocities $u_0$ (Table 5). The imaging reveals that the free-flight distances before first collisions increase with increasing surface slope. The particles then experience complex collisions with the surface that randomize their motions, including the onset of rotation.

For all surface slopes, the large rounded particles systematically travel farther than the large angular particles, and the small particles typically travel distances similar to those of the large angular particles (Figure 11). The data are reasonably fit by the theoretical curves, notably at small and large $Ki$. At intermediate $Ki$, several cases involve a systematic deviation from the curves. Estimated parametric values are provided in Table 6, where estimates of the variability in $\mu$, $\alpha$, $Ki$ and $Ki_*$ are based on a Monte Carlo analysis as described in Appendix C. As with the data of Gabet and Mendoza (2012), we displace





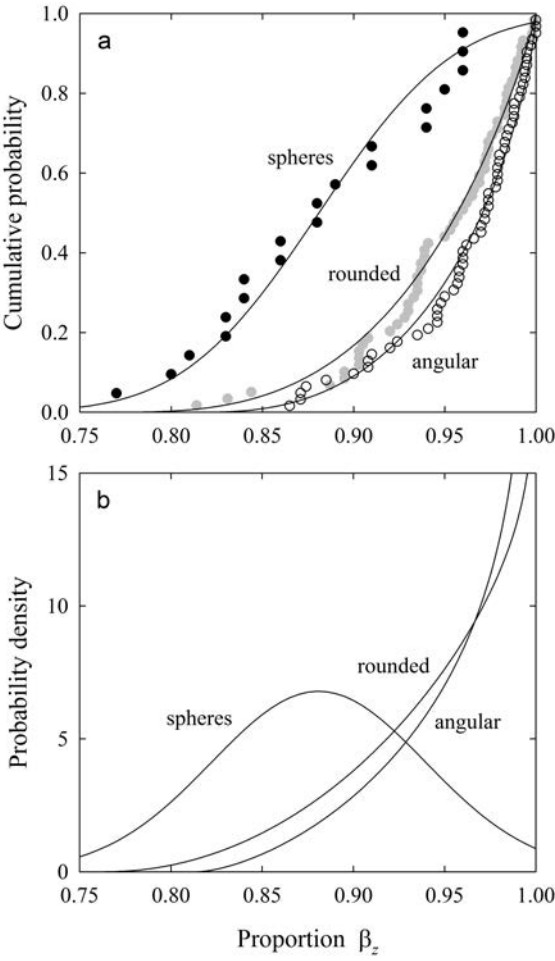

**Figure 10.** Plot of (a) cumulative distributions of $\beta_z$ for glass spheres fit to a Gaussian distribution and rounded and angular particles fit to a beta distribution, and (b) associated probability density functions of fitted distributions. Collisions are on rough surface.

the starting position ($x = 0$) to the first inflections in the raw exceedance probability plots, then recalculate distances and exceedance probabilities. A specific example is provided in Appendix B.

Note that, for reference below, two value of $\mu$, $\alpha$, $Ki$ and $Ki_*$ are provided. The first value is based on the measured launch velocity $u_0$ (Table 5). The second value is based on a reduced velocity. Namely, we do not know the average energy state of the particles at the truncation position used in the fitting of $A$ and $B$, although it most likely is smaller than that associated with the launch velocity. To provide a sense of the uncertainty in the calculated values we thus assume that the particle energy is reduced by half of its initial launch energy, although this is less likely with increasing surface slope and gravitational heating. At small slopes $S$ this adjustment has a larger effect on $\mu$ than on $\alpha$. At large slopes this adjustment has a larger effect on $\alpha$.



**Table 5.** Slope-parallel launch velocities $u_0$ measured from high-speed imaging.

| Particles | Slope | $u_0$ (m s$^{-1}$) | $N$ |
|---|---|---|---|
| angular | 0.00 | $0.58 \pm 0.036$ | 10 |
| | 0.09 | $0.79 \pm 0.053$ | 5 |
| | 0.15 | $0.86 \pm 0.041$ | 5 |
| | 0.18 | $0.84 \pm 0.084$ | 4 |
| | 0.25 | $1.00 \pm 0.028$ | 4 |
| | 0.28 | $0.98 \pm 0.036$ | 5 |
| rounded | 0.00 | $0.60 \pm 0.060$ | 10 |
| | 0.09 | $0.80 \pm 0.021$ | 5 |
| | 0.15 | $0.86 \pm 0.050$ | 5 |
| | 0.18 | $0.88 \pm 0.027$ | 5 |
| small | 0.00 | $0.55 \pm 0.15$ | 10 |
| | 0.09 | $0.77 \pm 0.038$ | 5 |
| | 0.15 | $0.89 \pm 0.038$ | 5 |
| | 0.18 | $0.91 \pm 0.011$ | 5 |
| | 0.25 | $0.97 \pm 0.013$ | 5 |

For the lower slopes ($S = 0$ to $S = 0.15$), all particles experience rapid thermal collapse ($A < 0$). However, between $S = 0.15$ and $S = 0.18$, the rounded particles appear to transition to a heavy-tailed behavior. By $S = 0.25$, no rounded particles stop on the surface; gravitational heating far exceeds frictional cooling. At this slope the angular and small particles exhibit heavy-tailed behavior with net heating. At $S = 0.28$, only angular particles stopped on the surface with net heating, and these are strongly censored (126 of 210 particles stopped).

For a given slope, values of the friction factor $\mu$ for the angular and small particles are systematically larger than values for the rounded particles. This result is consistent with the expectation that angular particles on average experience a greater energy loss during collisions than rounded particles, and that larger rounded particles are less likely than are small particles (both rounded and angular) to be influenced by the surface roughness texture during collisions. Whereas values of the factor $\alpha$ generally increase with slope $S$ (and Kirkby number $Ki$), no pronounced differences between particle size or shape appear.

Video and audio recordings (Supplementary Materials, Vanderbilt University Institutional Repository, https://ir.vanderbilt.edu/handle/1803/9742) provide clear evidence in support of the results above. In particular the files "Rounded_colinear.avi" and "Angular_colinear.avi" show examples of colinear collisions on the hard slate, with negligible rotation following collision and maximum vertical rebound. These examples are used in estimating the coefficient of restitution $\epsilon$ for the particle-slate collisions. The angular particle collision is a low probability event (dubbed the "pogo stick" by M. Schmeeckle) in that the point of impact involves a particle corner that becomes aligned directly beneath the center of mass at the instant of impact.





**Table 6.** Fitted and estimated values of the parameters for the data shown in Figure 11 with coefficients of variation in parentheses.

| Particles | Slope $S$ | $A$ | $B$ (m) | $Ki$ | $Ki_*$ | $\mu$ | $\alpha$ | $\mu_x^a$ (m) | $\mu_x^b$ (m) |
|---|---|---|---|---|---|---|---|---|---|
| Angular | 0 | -0.54 | 0.033 | 0.00 (—)$^c$ | 0.62 (0.026) | 0.46 (0.15) | 1.31 (0.043) | 0.021 | 0.022 |
| | | | | 0.00 (—) | 0.62 (0.026) | 0.23 (0.19) | 1.31 (0.043) | | |
| | 0.09 | -0.39 | 0.075 | 0.23 (0.15) | 0.64 (0.028) | 0.40 (0.12) | 1.40 (0.069) | 0.054 | 0.063 |
| | | | | 0.37 (0.14) | 0.71 (0.034) | 0.24 (0.14) | 1.72 (0.15) | | |
| | 0.15 | -0.36 | 0.12 | 0.40 (0.075) | 0.71 (0.021) | 0.37 (0.076) | 1.73 (0.054) | 0.088 | 0.093 |
| | | | | 0.58 (0.067) | 0.79 (0.024) | 0.26 (0.069) | 2.44 (0.094) | | |
| | 0.18 | -0.56 | 0.22 | 0.55 (0.098) | 0.83 (0.024) | 0.33 (0.10) | 2.96 (0.13) | 0.14 | 0.15 |
| | | | | 0.71 (0.085) | 0.89 (0.025) | 0.25 (0.088) | 4.59 (0.26) | | |
| | 0.25 | 0.30 | 0.35 | 0.98 (0.012) | 0.80 (0.021) | 0.26 (0.012) | 2.55 (0.086) | 0.50 | 0.40 |
| | | | | 0.99 (0.0062) | 0.90 (0.011) | 0.25 (0.0060) | 5.04 (0.11) | | |
| | 0.28 | 0.77 | 1.18 | 1.07 (0.012) | 0.95 (0.0064) | 0.26 (0.012) | 9.29 (0.11) | 5.13 | 0.58 |
| | | | | 1.03 (0.0062) | 0.97 (—) | 0.27 (0.0063) | 19.2 (0.13) | | |
| Rounded | 0 | -0.51 | 0.049 | 0.00 (—) | 0.60 (0.028) | 0.32 (0.22) | 1.27 (0.042) | 0.033 | 0.034 |
| | | | | 0.00 (—) | 0.60 (0.028) | 0.16 (0.30) | 1.27 (0.042) | | |
| | 0.09 | -0.24 | 0.119 | 0.36 (0.058) | 0.63 (0.021) | 0.25 (0.059) | 1.35 (0.036) | 0.096 | 0.101 |
| | | | | 0.53 (0.049) | 0.73 (0.021) | 0.12 (0.049) | 1.84 (0.056) | | |
| | 0.15 | 0.10 | 0.19 | 0.76 (0.033) | 0.66 (0.053) | 0.20 (0.033) | 1.47 (0.11) | 0.21 | 0.21 |
| | | | | 0.86 (0.024) | 0.81 (0.037) | 0.17 (0.025) | 2.59 (0.16) | | |
| | 0.18 | 0.020 | 0.28 | 0.80 (0.018) | 0.79 (0.020) | 0.22 (0.019) | 2.35 (0.073) | 0.29 | 0.30 |
| | | | | 0.89 (0.012) | 0.88 (0.013) | 0.20 (0.013) | 4.24 (0.10) | | |
| Small | 0 | -0.49 | 0.035 | 0.00 (—) | 0.60 (0.030) | 0.36 (0.57) | 1.24 (0.044) | 0.24 | 0.27 |
| | | | | 0.00 (—) | 0.60 (0.030) | 0.18 (0.82) | 1.24 (0.043) | | |
| | 0.09 | -0.51 | 0.098 | 0.26 (0.095) | 0.71 (0.018) | 0.35 (0.096) | 1.70 (0.044) | 0.065 | 0.072 |
| | | | | 0.41 (0.095) | 0.77 (0.021) | 0.22 (0.096) | 2.14 (0.071) | | |
| | 0.15 | -0.35 | 0.12 | 0.39 (0.071) | 0.70 (0.021) | 0.38 (0.072) | 1.69 (0.049) | 0.089 | 0.097 |
| | | | | 0.56 (0.064) | 0.79 (0.022) | 0.27 (0.065) | 2.35 (0.083) | | |
| | 0.18 | -0.51 | 0.24 | 0.54 (0.039) | 0.82 (0.0098) | 0.33 (0.039) | 2.77 (0.044) | 0.16 | 0.17 |
| | | | | 0.70 (0.026) | 0.88 (0.0075) | 0.26 (0.027) | 4.28 (0.056) | | |
| | 0.25 | 0.30 | 0.41 | 0.98 (0.010) | 0.84 (0.014) | 0.25 (0.010) | 3.16 (0.075) | 0.59 | 0.43 |
| | | | | 0.99 (0.0051) | 0.92 (0.0069) | 0.25 (0.0051) | 6.26 (0.078) | | |

$^a$ Estimated from parameters $A$ and $B$.

$^b$ Estimated from data, recognizing that these do not incorporate distances of particles that moved beyond measurement distance of 1.9 m.

$^c$ Notation $(-)$ means undefined or coefficient of variation is less than 0.01.

One of the more compelling results appearing in several of the videos is when the translational kinetic energy of a particle at first impact is converted to translational kinetic energy involving transverse motion and rotational kinetic energy, then

during a second or third collision, rotational energy is converted back to vertical motion thence to gravitational potential en-
ergy. The likelihood of this occurring increases with particle angularity, where noncollinear collisions are the rule rather than
exception, and pointy particle corners lead to unusual collision configurations. The file "Angular_all_rotational.avi" shows
a particularly strong conversion of translational to rotational motion with the initial collision on hard slate. The file "Angu-
lar_rotational_to_vertical.avi" shows conversion of rotational to vertical motion with the second collision. The file "Semian-
gular_rotational_die.avi" shows rapid cessation of motion following conversion of rotational to vertical motion.

The geometry of a noncollinear collision following the vertical drop of an angular particle is different from that of a particle
at relatively small incident angle. Nonetheless, the strong conversion of translational to rotational motion associated with the
former is analogous to the behavior of a particle that experiences stick during a small incident angle collision with conversion
of translational to rotational energy (Appendix E in Furbish et al., 2020a).

The files "Angular_18%slope.avi" and "Angular_28%slope.avi" show examples of angular particles launched from the cata-
pult onto the rough surface. Although the surfaces in these videos appear flat because of camera alignment, the slopes are $S =
0.18$ (10.2 degrees) and $S = 0.28$ (15.6 degrees), so gravitational heating starts immediately. The particles are launched with
negligible initial rotation and the motions start to become randomized, including the onset of rotation, following free flight and
initial surface collisions. Rather than decelerating, gravitational heating maintains velocities similar to the launch velocities. In-
deed, the particle on $S = 0.18$ seems likely to stop, but then continues with heating. For contrast the file "Rounded_0slope.avi"
shows an example of a rounded particle that rapidly decelerates then "dies" when launched onto the flat rough surface ($S = 0$).
The increase in free-flight distances (before initial collisions) with increasing slope are apparent in the three videos.

Audio recordings of particle-surface interactions during their downslope motions reveal the distinctive clickety-click sounds
of collisions ("Bouncing.m4a"), which are markedly different from the sounds emitted by particles that are either gently
or forcefully made to slide on the rough surface ("Sliding.m4a"). These clickety-click sounds occur with high frequency,
particularly when particles are in a tumbling (nominally "rolling") mode, giving way to decreasing frequency when particles
undergo runaway bouncing motions. In contrast, sliding motions emit continuous scraping sounds. The key result of these
recordings is to audibly reinforce the idea that motions involve collisional friction rather than a sliding Coulomb-like behavior,
except in association with the brief collision impulses as described in collision mechanics theory (Brach, 1991; Stronge, 2000).

Further analyses of these detailed particle motions in relation to downslope and cross-slope motions are to be reported
elsewhere.

## 4   Field measurements

### 4.1   DeBiase et al. (2017)

#### 4.1.1   Experiments

The field-based experiments reported by DiBiase et al. (2017) involved launching three different sizes of particles down a
natural hillslope surface. The particle size classes included diameters $D = 2-3$ cm, $D = 4-6$ cm and $D = 9-12$ cm, involving





58, 93 and 43 particles, respectively. Of these, 53, 61 and 14 particles stopped within a 14 m measurement distance with approximately uniform steepness of 38 degrees. The distributions of travel distance systematically varied with the different particle sizes. Further details of the experiments, including measurements of surface roughness, are provided by DiBiase et al. (2017).

As with the data of Gabet and Mendoza (2012), we again fit the parameters $A$ and $B$, then calculate values of $\mu$ and $\alpha$ assuming a value of $\gamma$. We also displace the starting position ($x = 0$) to the first inflections in the raw exceedance probability plots for the smaller two particle sizes, then recalculate distances and exceedance probabilities.

### 4.1.2 Results

The data are reasonably well fit by the theoretical curves (Figure 12). Estimated parametric values are provided in Table 7, where estimates of the variability in $\mu$, $\alpha$, $Ki$ and $Ki_*$ are based on a Monte Carlo analysis as described in Appendix C.

**Table 7.** Fitted and estimated values of the parameters for the data shown in Figure 12 with coefficients of variation in parentheses.

| Particle size (m) | $A$ | $B$ (m) | $Ki$ | $Ki_*$ | $\mu$ | $\alpha$ | $\mu_x^a$ (m) | $\mu_x^b$ (m) |
|---|---|---|---|---|---|---|---|---|
| 0.025 | 0.81 | 2.4 | 1.04 (0.013) | 0.97 (—)$^c$ | 0.76 (0.012) | 16.4 (0.14) | 12.6 | 3.32 |
| 0.05 | 1.7 | 5.1 | 1.01 (0.015) | 1.00 (—) | 0.78 (0.013) | 227 (0.013) | — | 4.04 |
| 0.10 | 5.0 | 8.8 | 1.00 (0.049) | 1.00 (—) | 0.79 (0.034) | 3160 (—) | — | 3.74 |

$^a$Estimated from parameters $A$ and $B$.

$^b$Estimated from data, recognizing that these do not incorporate distances of particles that moved beyond measurement distance of 14 m.

$^c$Notation (—) means coefficient of variation is less than 0.01.

For all particle diameters the Kirkby number $Ki \approx 1$, and the fits are insensitive to the value of $\gamma$. Moreover, estimated values of the friction factor $\mu$ are similar; these do not show a systematic change with particle size. The estimated values of $A$ suggest that the smallest particles represent a distribution with finite mean but undefined variance ($1/2 \leq A < 1$). The larger two particle sizes represent conditions with an undefined mean and undefined variance ($A \geq 1$).

In contrast to the ambiguity of an exponential versus a Pareto fit to the data of Gabet and Mendoza (2012) for $A \geq 0$ (Figure 4b), the concavity in the semi-log exceedance probability plot (Figure 12) for the smaller two particle sizes is readily apparent and consistent with a Pareto distribution. Certainly one could fit a straight line to these data, but the fit would degrade (as revealed, although not shown, by quartile-quartile plots). Nonetheless, we must be cautious. Inasmuch as the theoretical distribution is the correct choice, then the data represent only a fraction of the total probability. For the smallest particles about 15% of the tail is not sampled. For the intermediate particles about 35% is not sampled; and for the largest particles about 70% of the tail is not sampled. This reinforces the difficulty of working with tail-censored data with relatively small sample sizes. Note also that for the smallest particle size the average travel distance $\mu_x$ calculated from the data is similar to the estimated parameter $B$ but is significantly smaller than the average estimated from the values of $A$ and $B$ (Table 7). This occurs because





the data "see" probability distributed in a manner that is not dissimilar from that expected for an exponential distribution, whereas the values of $A$ and $B$ incorporate information contained in the tail of the Pareto distribution.

Based on reported particle velocities, values of the initial average kinetic energies $E_{a0}$ of the three particle sizes are $1.0 \times 10^{-5}$ J, $1.1 \times 10^{-4}$ J and $8.5 \times 10^{-4}$ J. Estimated values of the average kinetic energies $E_a$ measured over the total travel distances are $8.0 \times 10^{-6}$ J, $1.1 \times 10^{-4}$ J and $2.2 \times 10^{-3}$ J. The change in average energies $\Delta E_a$ are $-2.5 \times 10^{-6}$ J, $0.0 \times 10^{0}$ J and $1.4 \times 10^{-3}$ J. These changes are qualitatively consistent with net cooling, isothermal conditions and net heating, although in all three cases the estimated parametric values suggest net heating ($Ki \geq Ki_*$). Using the largest value of the friction factor

$\mu$ (Table 7), the Kirkby number of the 40 degree surface immediately downslope from the measurement site is $Ki \approx 1.1$. The transition Kirkby number $Ki_* = 1$. If particles on average experienced net heating on the upper 38 degree surface, then this result is consistent with the reported observation that particles reaching the steeper slope continued to the base of the hillslope without stopping.

## 4.2    Roth et al. (2020)

### 4.2.1    Experiments

The field-based experiments reported by Roth et al. (2020) involved dropping three different sizes of particles on eight natural hillslope surfaces in the Oregon Coast Range. Five of the hillslopes were covered with natural vegetation (designated by V), and included slope angles of zero, 14, 20, 24 and 39 degrees. Three of the hillslopes had recently been burned (designated by B) and included slope angles of 17, 20 and 28 degrees. Particle size classes involved average diameters of 1.7, 4.5 and 7.3 cm.

These were dropped from a height of $h \approx 0.2$ m onto each hillslope surface. The distributions of travel distances systematically varied with slope angle, particle size and surface roughness conditions.

The surfaces of the vegetated hillslopes had a layer of duff, woody debris and banana slugs beneath small plants (e.g., ferns) and trees. The surfaces of the burned hillslopes had little vegetal cover and were markedly smoother than the vegetated sites. Further details of the experiments, including measurements of surface roughness, are provided by Roth et al. (2020).

Banana slugs, whose locomotive energetic costs are constrained by the shear-thinning rheology of their mucus (Lauga and Hosoi, 2006), appear as slow moving Dirac functions in the power spectra of surface elevation. None were injured during the experiments.

As above, we fit the parameters $A$ and $B$, then calculate values of $\mu$ and $\alpha$ assuming a value of $\gamma$. We also displace the starting position ($x = 0$) to the first inflection in the raw exceedance probability plots, then recalculate distances and exceedance probabilities. This displacement is applied only for cases involving relatively small travel distances (typically the smallest particles), and is only about one or two particle diameters. For travel distances involving tens of m, we focus the fitting on the

central part of the data, deemphasizing the fit for small values and for the extreme tails. In addition, changes in surface slope occur on all sites, and we restrict the data fitting to positions upslope of these changes. These changes occur at: 2.7 m (V14), 3.5 m (V20), 11 m (V24), 16.6 m (V39), 34 m (B17), 31 m (B20) and 33 m (B28). For site V0, 20 travel distances were measured for each particle size class. Initial examination indicated that the distributions of travel distances were similar, so we pooled





these data. By dropping (rather than launching) the particles, initial energies are less certain than those in the experiments

reported above. We calculate the impact velocities, assume a coefficient of restitution, then use the average downslope reflection

velocities. We note that uncertainty in the initial energies affects the estimates of the quantities $\mu$ and $\alpha$ but does not influence

the values of the parameters $A$ and $B$ obtained from the data fitting.

### 4.2.2   Results

The data for the V sites are reasonably well fit by the theoretical curves (Figure 13). In these two examples as well as in

cases not plotted, the estimated parameter $A$ systematically increases with particle size (Table 8), and reflects a transition from

**Table 8.** Fitted and estimated values of the parameters for the data reported by Roth et al. (2020) as shown in Figure 13 and Figure 14 with
coefficients of variation in parentheses.

| Site | Slope (deg) | Particle size (m) | $A$ | $B$ (m) | $Ki$ | $Ki_*$ | $\mu$ | $\alpha$ | $\mu_x^a$ (m) | $\mu_x^b$ (m) |
|------|------|------|------|------|------|------|------|------|------|------|
| V | 0 | all | -0.41 | 0.087 | 0.00 (—)$^c$ | — | — | — | 0.06 | 0.06 |
|  | 14 | 0.017 | -0.41 | 0.165 | 0.95 (0.0061) | 0.98 (—) | 0.26 (0.0062) | 21.6 (0.095) | 0.12 | 0.13 |
|  | 14 | 0.045 | 0.45 | 0.23 | 1.01 (—) | 0.98 (—) | 0.25 (—) | 28.3 (0.10) | 0.42 | 0.33 |
|  | 14 | 0.073 | 1.1 | 0.13 | 1.08 (0.014) | 0.97 (—) | 0.23 (0.014) | 15.0 (0.11) | — | 0.34 |
|  | 20 | 0.017 | -0.23 | 0.72 | 0.99 (—) | 0.99 (—) | 0.37 (—) | 64.0 (0.097) | 0.59 | 0.61 |
|  | 20 | 0.045 | -0.30 | 1.8 | 1.00 (—) | 1.00 (—) | 0.37 (—) | 159 (0.086) | 1.38 | 1.34 |
|  | 20 | 0.073 | 0.20 | 1.0 | 0.99 (—) | 0.99 (—) | 0.36 (—) | 87.9 (0.10) | 1.25 | 1.25 |
|  | 24 | 0.017 | -0.06 | 0.60 | 1.00 (—) | 1.00 (—) | 0.45 (—) | 44.8 (0.097) | 0.57 | 0.58 |
|  | 24 | 0.045 | -0.01 | 2.3 | 1.00 (—) | 1.00 (—) | 0.45 (—) | 170 (0.099) | 2.28 | 2.38 |
|  | 24 | 0.073 | 0.01 | 3.4 | 1.00 (—) | 1.00 (—) | 0.45 (—) | 251 (0.10) | 3.43 | 3.35 |
|  | 39 | 0.045 | -0.12 | 0.30 | 0.95 (—) | 0.97 (—) | 0.85 (—) | 15.0 (0.093) | 0.27 | 0.30 |
|  | 39 | 0.017 | -0.38 | 3.7 | 0.99 (—) | 1.00 (—) | 0.81 (—) | 177 (0.098) | 2.68 | 2.68 |
|  | 39 | 0.073 | 0.70 | 4.8 | 1.00 (—) | 1.00 (—) | 0.81 (—) | 228 (0.10) | 16.0 | 5.25 |
| B | 17 | 0.017 | -0.39 | 0.27 | 0.96 (—) | 0.98 (—) | 0.32 (—) | 28.8 (0.095) | 0.19 | 0.20 |
|  | 17 | 0.045 | -0.03 | 0.49 | 0.99 (—) | 0.99 (—) | 0.31 (—) | 50.8 (0.10) | 0.48 | 0.83 |
|  | 17 | 0.073 | 0.67 | 0.39 | 1.01 (—) | 0.99 (—) | 0.30 (—) | 39.5 (0.10) | 1.18 | 1.41 |
|  | 20 | 0.017 | 0.10 | 0.18 | 0.98 (—) | 0.97 (—) | 0.37 (—) | 16.1 (0.099) | 0.20 | 0.22 |
|  | 20 | 0.045 | 1.30 | 0.90 | 1.02 (—) | 0.99 (—) | 0.36 (—) | 77.5 (0.099) | — | 4.01 |
|  | 20 | 0.073 | 1.68 | 0.64 | 1.04 (0.0060) | 0.99 (—) | 0.35 (0.0060) | 54.1 (0.10) | — | 2.97 |

$^a$ Estimated from parameters $A$ and $B$.

$^b$ Estimated from data, recognizing that these do not incorporate distances of particles that moved beyond positions of noted slope changes.

$^c$ Notation $(-)$ means undefined or coefficient of variation is less than 0.01.

rapid thermal collapse ($A < 0$) to approximately isothermal conditions ($A \approx 0$) or net heating ($A > 0$). Interestingly, travel

distances on the steep V24 slope are systematically larger than on V14. Yet evidently the surface roughness on V24 leads to





approximately isothermal conditions for the larger particles, whereas the roughness on V14 leads to net particle heating. The

data for the B sites (Figure 14) similarly show that the estimated parameter $A$ systematically increases with particle size (Table 8), transitioning from rapid thermal collapse to net heating. Note that the fitted shape and scale parametric, $A$ and $B$ (Table 8), are consistent in sign and approximate magnitude with those presented by Roth et al. (2020) for the same data set. This paper also presents exceedance probability plots for all 21 experiments (excluding the zero slope case). Also note that estimates of the variability in $\mu$, $\alpha$, $Ki$ and $Ki_*$ are based on a Monte Carlo analysis as described in Appendix C.

The steepest burned site, B28, offers a further interesting perspective on particle motions. On this steep and relatively smooth

hillslope, exceedance probabilities associated with all three particle sizes cannot be fit with reasonable fidelity by individual curves. Rather, the data suggest a mingling of two particle behaviors — rapid cooling for many particles, and runaway heating for a second group leading to a pronounced heavy tail (Figure 15) — in effect giving a mixed distribution. Namely, let $x_1$ and $x_2$ denote the travel distances of the two groups, and let $p_1$ denote the proportion represented by first group such that $p_2 = 1 - p_1$ is the proportion of the second group. The simplest form of a mixed distribution is

$$f_x(x) = p_1 f_{x_1}(x_1) + (1 - p_1) f_{x_2}(x_2) \tag{39}$$

and the cumulative distribution is

$$F_x(x) = p_1 \int_0^x f_{x_1}(x_1') \, dx_1' + (1 - p_1) \int_0^x f_{x_2}(x_2') \, dx_2' \tag{40}$$

where primes denotes variables of integration. As above, the exceedance probability is $R_x(x) = 1 - F_x(x)$.

For the three particle sizes, exceedance probabilities are well matched by the weighted sum of a nearly exponential distribution ($A_1 \approx 0$) reflecting isothermal conditions and a heavy-tailed distribution ($A_2 > 0$) reflecting net heating (Table 9). Note,

**Table 9.** Fitted and estimated values of the parameters for the data shown in Figure 15.

| Site | Slope (degrees) | Particle size (m) | $A_1$ | $B_1$ (m) | $A_2$ | $B_2$ (m) | $p_1$ |
|---|---|---|---|---|---|---|---|
| Burned | 28 | 0.017 | 0.001 | 0.50 | 0.70 | 2.0 | 0.85 |
| | 28 | 0.045 | 0.30 | 0.90 | 8.2 | 8.0 | 0.38 |
| | 28 | 0.017 | 0.01 | 0.50 | 2.9 | 110 | 0.34 |

however, that the parametric values $A_1$, $B_1$, $A_2$ and $B_2$ cannot be combined to estimate associated factors such as $\mu$ and $\alpha$. Although in these three experiments (B28) the particles in each size group are nominally similar, we nonetheless suspect that the steep slope and relatively smooth surface give conditions that "filter" the particles into two subsets. One subset consists

of particles whose motions are strongly randomized and become disentrained over short distances. The other subset consists of particles whose motions by chance quickly transition to rotation and travel much longer distances over the smooth surface. This filtering likely includes effects of the dropping of non-spherical particles. Namely, particles that are by chance initially dropped onto their relatively flat faces are less likely to transition to rotation and thus are more likely to travel short distances.





We also suspect the existence of a similarly nonuniform behavior in the Vanderbilt data, manifest as systematic variations in several of the exceedance probability plots (Figure 11).

Using the same data set, Roth et al. (2020) directly calculate the disentrainment rate function $P_x(x)$ using finite-differencing of the empirical cumulative distribution and exceedance probability functions. Although noisy, the data clearly illustrate the forms of $P_x(x)$ representing rapid thermal collapse ($A < 0$), approximately isothermal conditions ($A \approx 0$) and net heating ($A > 0$). Of particular note is the result that the roughness of natural vegetation exerts a strong cooling effect, and that the spatial structure of roughness elements together with local changes in surface slope can contribute to noticeable variations in travel distances about those expected for nominally uniform conditions.

## 5 Analysis

We emphasize at the outset a key point in comparing travel distances measured in experiments (laboratory or field) with theoretical distributions. By definition a sample of measured values drawn from a distribution possesses a finite sample mean and variance, regardless of the form of the underlying distribution. If the underlying distribution possesses a finite mean and variance (e.g., an exponential distribution), then the calculated sample average and variance are unbiased estimates of the underlying parametric values. If the mean or variance of the underlying distribution is undefined (e.g., the generalized Pareto distribution for $A \geq 1/2$), then the calculated sample average and variance have no meaningful relation to the underlying (undefined) moments. We can never know this, although it might be suggested, for example, by the absence of convergence of estimated moments with increasing sample size or from multiple samples. The best we can do is to infer the veracity of the form of the distribution from descriptive statistics (e.g., exceedance probability plots, quartile-quartile plots), but this generally requires large data sets to support the tails of heavy-tailed distributions. In some of the comparisons above, there is the real possibility that calculated averages are just numbers associated with a distribution whose mean is undefined, such that the calculated values do not meaningfully characterize a property (e.g., absence of central tendencies) of the underlying distribution.

In contrast, estimates of the parameters $A$ and $B$ are less sensitive to this uncertainty when these values are used to calculate moments (if they exist) — but only if the selected form of the distribution is the correct choice. The mechanical basis of the generalized Pareto distribution lends confidence, but does not guarantee, that it is the correct choice. Moreover, uncertainty in the estimated values of $A$ and $B$ increases when a decreasing proportion of the tail of the distribution is sampled. Indeed, one can never know the form of the censored tail (Balio et al., 2019).

With these points in mind, we suggest that the fits presented above are consistent with the idea that each of the data sets represents a specific case of the generalized Pareto distribution. To further illustrate this idea we calculate the following quantities:

$$R_* = R_x^A \qquad \text{and} \qquad x_* = \frac{A}{B}x + 1. \tag{41}$$

Based on Eq. (14), values of the modified exceedance probability $R_*$ and the dimensionless travel distance $x_*$ should collapse to a single straight line in a log-log plot with slope of $-1$ (Figure 16). Note that these plots magnify the deviations in the tails





of the distribution. Also note that these fits suggest that all data, if plotted together, would collapse to the same line spanning more than three orders of magnitude of the dimensionless travel distance $x_*$. This does not prove, but nonetheless supports, the idea that the generalized Pareto distribution correctly describes the energetics of the behavior of rarefied particle motions for a variety of slope and surface roughness conditions.

The bounded form of the generalized Pareto distribution ($A < 0$) must not be viewed as involving a "hard" boundary. Because of the stochasticity of motions associated with varying sizes and shapes, some particles by chance "leak" beyond the position $x = B/|A|$. This aspect of the formulation is necessarily simplified. What is clear, nonetheless, is the rapid thermal collapse reflected by the (approximately) bounded form of the distribution in the laboratory measurements of Gabet and Mendoza (2012) (Figure 3, Figure 4) and the measurements at Vanderbilt (Figure 11), and the field-based measurements of Roth et al. (2020) (Figure 13, Figure 14).

From an empirical point of view the data are consistent with the generalized Pareto distribution, and reflect the predicted variation in behavior from rapid thermal collapse to approximately isothermal conditions to net heating of particles. Nonetheless we proceed by asking whether the estimated values of the quantities $\mu$ and $\alpha$ make physical sense, while recognizing that these quantities do not readily map to established formulations of friction, and represent a complexity that cannot be encapsulated in idealized collision mechanics (Appendix E in Furbish et al., 2020a).

The laboratory measurements with zero slope merit particular attention. The Kirkby number $Ki$ is known and zero. The effect of heating, and thus the influence of heating on $\alpha$, is removed. Focusing on the length scale $L_c$ given by Eq. (37), motions are mass independent and the initial velocity is approximately fixed. Assuming fixed $\gamma$ and comparing the angular and rounded particles in the Vanderbilt data (Table 6), the effect of particle angularity evidently appears as a difference in $\mu$, consistent with $\mu \sim \langle \beta_x \rangle$ and the measurements indicating that angular particles on average extract more translational energy during collisions than rounded particles (Figure 9, Figure 10). This also is consistent with the measurements of Gabet and Mendoza (2012) for a flat surface in that all values of $\mu$ in the Vanderbilt data are larger than the value of $\mu$ for a spherical particle in the Gabet and Mendoza data. We suspect that the small particles "feel" the roughness texture more than the larger rounded and angular particles; and because the small particles are a mixture of rounded and angular shapes, the value of $\mu$ is similar to the angular particles. These differences in the values of $\mu$ persist with larger slopes, where the values of $\mu$ for rounded particles remains less than the other values (although we must be cautious to not overinterpret these differences given the uncertainty of the calculations). At zero slope the values of $\alpha$ are similar across particle shape and size. No similar systematic variation in $\alpha$ with particle size and shape is apparent, although rounded particles appear to be more readily heated with increasing slope.

The field-based measurements of DiBiase et al. (2017, Table 7) and Roth et al. (2020, Table 8) suggest that the friction factor $\mu$ is insensitive to particle size for the same slope and roughness conditions, and these data together with the laboratory measurements of Gabet and Mendoza (2012, Table 8) and the Vanderbilt data suggest that $\mu$ systematically varies with surface slope $S$ (Figure 17). Note that the Vanderbilt data in this figure are based on the reduced velocity calculations (Table 6). For completeness this figure is reproduced in Appendix E using the initial launch velocities $u_0$ (Table 5).

Values of $\mu$ for $Ki < 1$ systematically fall above the 1:1 line (Figure 17), then converge to this line as $Ki \to 1$. Using Eq. (32) to estimate $\mu$, evidently near $S = 0$ the second term on the right side of this equation dominates and gives positive $\mu$ with





negative $A$ for the smallest four slope angles in the data of Gabet and Mendoza (2012, Table 2) and the Vanderbilt data (Table
6). The magnitude of this term then decreases (for $A > 0$) with increasing $S$ such that $\mu \sim S$ as $Ki \to 1$ according to Eq. (8).
Note that Eq. (32) does not provide a physical explanation of the factor $\mu$; it is just an estimate of $\mu$ based on the parameters $A$
and $B$.

As summarized in Section 2.4, scaling suggests that the factor $\mu \sim M(\theta)$ is independent of particle size (Furbish et al.,
2020a). This consistency with the experimental data reinforces the idea that the elements of $\mu \sim M(\theta)$, despite the complexity
of the collisions involved, are akin to results from idealized collision mechanics, namely, that these elements are determined
by the coefficients associated with tangential and normal impulses, where particle size is not involved for a given slope and
surface roughness. Recall that the expected dependency $\mu \sim M(\theta)$ on the slope angle $\theta$ arises because the expected surface
normal impact velocity varies with this angle (Appendix E in Furbish et al., 2020a). However, this probably is insufficient to
explain the relation in Figure 17. Unfortunately, we cannot further unfold the physical basis of $\mu$ in relation to particle-surface
interactions beyond observing that the values of $A$ and $B$ return estimates of $\mu$ that are consistent with the expectation of its
slope dependency, independent of particle size. On empirical grounds, as $Ki \to 1$ the factor $\mu \to S$ reflects an approximate
balance between heating and cooling with respect to translational energy. That is, the rate of extraction of translational energy
(partitioned to all other forms) increases to match the rate of heating. Presumably this balance is exceeded ($Ki \gg 1$) with
slopes that are so steep that cooling is insufficient for deposition to occur — as in several of the experiments described above.
Turning to the quantity $\alpha$, we plot the estimated values of this factor based on Eq. (33) versus the Kirkby number $Ki$
(Figure 18) together with the function in Eq. (35). Note that the Vanderbilt data in this figure are based on the reduced velocity
calculations (Table 6). For completeness this figure is reproduced in Appendix E using the initial launch velocities $u_0$ (Table
5).

The data from Gabet and Mendoza (2012) support the idea that $\alpha$ systematically increases with $Ki$ and becomes unbounded
near $Ki \sim 1$. The Vanderbilt data similarly support this idea. The data for all three particle sizes from DiBiase et al (2017)
involve $Ki \approx 1$ and thus only reinforce the unbounded behavior of $\alpha$. Similarly, the data from Roth et al. (2020) support this
behavior. Note that values with large $Ki$ and $A \geq 1$ are not meaningful, as the underlying deposition length scales $L_c$ are
undefined. Also note that because the Kirkby number $Ki$ and the factor $\mu$ are specified in the fits of the data from Kirkby and
Statham (1975, Figure 6) rather than being estimated from $A$ and $B^*$, we do not plot these values in Figure 18.

Recall that $\alpha$ reflects a direct effect of heating, namely, to decrease the likelihood of deposition by decreasing the proportion
of particles that cool to sufficiently low energies for deposition to occur. This translates to suppressing the disentrainment rate
and increasing the deposition length scale $L_c$, rewritten here as

$$L_c = \frac{\alpha E_a}{\gamma m g \mu \cos \theta} = \frac{\alpha_0 E_a}{\gamma m g \mu \cos \theta (1 - \mu_1 Ki)} \quad (42)$$

With $E_a = (m/2)\langle u_p^2 \rangle$, the effect of particle mass $m$ does not explicitly appear. This means that any effect of particle size must
appear in $\alpha$ or $\mu$, or both. Similarly, any effect of particle angularity must appear in one or both of these quantities. The results
above, with particular reference to the flat rough surface in the Vanderbilt experiments, suggest that effects of angularity appear
in $\mu$, whereas the data sets together suggest that effects of particle size primarily appear in $\alpha$. We emphasize that the functional





relation of $\alpha$ to $Ki$ given by Eq. (35) is not definitive. Other functional forms are possible, although the basic form of Eq. (35)
seems to be reasonably consistent with the data (Figure 18). Although not explicit in the formulation, we suspect that the effect
of heating includes an increasing partitioning of energy into rotational motion that is amplified for larger particles for a given
slope and roughness, giving a decreasing likelihood of stopping as reflected in increasing $\alpha$. Further disentangling the effects
of $\alpha$ and $\mu$ must a await clearer mechanical basis for these quantities.

To reinforce the idea of a mixed distribution, consider the example of angular and rounded particles from the Vanderbilt
experiments for $S = 0$ (Figure 11a). When pooled these data can be approximately fitted (not shown) to a generalized Pareto
distribution. However, the data are well fit using the mixed distribution defined by Eq. (39) (Figure 19). Note that this mixture of
generalized Pareto distributions is not a generalized Pareto distribution. The distribution of travel distances of a mixture of par-
ticle sizes and shapes therefore must be described empirically or formed as a weighted mixture of distributions characterizing
the behavior of the individual particle size and shape groups involved.

## 6   Discussion and conclusions

The laboratory and field-based measurements of particle travel distances presented above provide clear evidence that these
distances are well described by a generalized Pareto distribution, where the form of the distribution reflects variations in particle
behavior associated with the balance between gravitational heating and frictional cooling by particle-surface collisions. These
behaviors vary from a bounded distribution associated with rapid thermal collapse to an exponential distribution representing
approximately isothermal conditions to a heavy-tailed distribution associated with net heating of particles. Here we reiterate a
point made in the first companion paper (Furbish et al., 2020a). Namely, we do not choose the generalized Pareto distribution
in the empirical manner of selecting a distribution based on goodness-of-fit criteria applied to data sets. Rather, this distribution
is dictated by the probabilistic physics of the problem, and is based on a description of the kinetic energy balance of a cohort
of particles treated as a rarefied granular gas, and a description of particle deposition that depends on the energy state of the
particles.

The experiments involving high-speed imaging of particle motions reinforce what we intuitively already understand. Relative
to a spherical particle, a rounded non-spherical particle is more likely to experience a noncollinear collision that converts the
translational energy of free fall into transverse motion and rotational energy; and an angular particle is more likely than is a
rounded particle to experience such conversions. The effect of this behavior is a systematic increase in the proportion $\beta_z$ with
increasing angularity. Moreover, following the first free fall collision, an angular particle is more likely than is a rounded particle
to experience a noncollinear collision that extracts either rotational or translational energy, or both. Translating this to surface-
parallel motions, a tumbling angular particle is more likely than is a rounded tumbling particle to experience a noncollinear
collision that extracts either translational energy or rotational energy, or both. Although we did not directly measure changes
in surface-parallel energy associated with collisions, we can infer that the proportion $\beta_x$ likely systematically increases with
increasing particle angularity as reflected in systematically shorter travel distances of angular particles relative to those of
rounded particles (Figure 11) on a surface with only a granular roughness texture. These experiments also illustrate the value





of treating $\beta_x$ as a random variable. Although this quantity is related to the normal coefficient of restitution $\epsilon$ as used in granular gas theory, the complexity and richness of collisions and associated conversions of energy among modes necessitates a probabilistic description in this problem.

The essence of rapid thermal collapse ($A < 0$) involves the situation in which gravitational heating is absent or is insufficient to replace frictional cooling, particularly with angular particles and small particles. That is, a small tumbling particle is more

likely than is a large tumbling particle to "see" the bumps and divots of the roughness texture at its scale, and to experience collisions that arrest its motion. Indeed, this is the basic lesson of experiments involving spheres rolling bumpety-bump over monolayer roughness elements (Dippel et al., 1997; Samson et al., 1998, 1999), the experiments of Kirkby and Statham (1975) involving particles moving down surfaces with different granular roughness, and the experiments of Roth et al. (2020) involving the different roughnesses of vegetated and burned hillslopes. With increasing gravitational heating the transition to a

heavy-tailed distribution of travel distances likely involves an increasing conversion of translational to rotational kinetic energy leading to larger travel distances with decreasing effectiveness of collisional friction. In this regard the analysis points to the need for further clarity concerning how particle size and shape in concert with surface roughness influence the extraction of particle energy and the likelihood of deposition.

Although not essential to the fitting of particle travel distances to the generalized Pareto distribution, it nonetheless is de-

sirable to have a clearer mechanical interpretation of the quantities $\mu$ and $\alpha$ and their relation to the Kirkby number $Ki$ in terms of particle properties and surface-roughness conditions, and the modes of particle motions. Here we note that the Pareto distribution with positive shape parameter $A$ can be obtained as a mixture of exponential distributions whose rate parameters are distributed as a gamma distribution (Appendix F). This result suggests an interesting physical interpretation of the Pareto distribution of particle travel distances, and it also may indicate a strategy for clarifying how particle size and shape in concert

with surface roughness influence the extraction of particle energy and the likelihood of deposition, inasmuch as the scale parameter $B$ is equivalent to the reciprocal of the expected disentrainment rate, $E(P_x)$. For example, Roth et al. (2020) show that $B$ systematically varies with particle size and surface slope based on the data described in Section 4.2.

We suggest that, in designing and conducting particle launching experiments, we have a propensity to select pretty particles, and rounded (if not spherical) particles are pretty. This is not a bad thing. But it skews our view of particle motions toward

the behavior of rounded particles. The experiments clearly demonstrate that particle angularity matters in the disentrainment process, specifically the likelihood of converting translational to rotational energy and the decreasing extraction of energy by collisional friction (Williams and Furbish, 2020).

Here in essence are the shortcomings of the formulation and its application to the experimental data sets of particle travel distances. We do not understand the transient (probabilistic) physics of near-launch conditions as particle motions become

randomized, so there is uncertainty in choosing the truncation distance and the associated particle energy state. Similarly, little is known about the distribution $f_{E_p}(E_p)$ of particle energy states $E_p$ and how this distribution might vary in the downslope direction. Thus, the assumption that the ratio $\gamma$ of the arithmetic and harmonic means of the particle energies remains fixed may be incorrect. This is a parsimonious choice to close the formulation analytically. The friction factor $\mu$ is tentative. Namely, its essential element, the expected proportion of energy extraction $\langle \beta_x \rangle$, is consistent with the experimental results as reflected





in the behavior of rounded versus angular particles, but the mechanical reasons for its asymptotic behavior, $\mu \to S$ as $Ki \to 1$ (Figure 17), remain unclear. Similarly the factor $\alpha$ is tentative. We need a clearer understanding of the elements that lead to increasing $\alpha$ and the associated lengthening of the deposition length scale $L_c$, notably in relation to particle size. Many of the individual fits between the data and the generalized Pareto distribution in the exceedance probability plots are reasonably close; but some exhibit systematic deviations about the fitted distribution. As is usual in this situation, it is difficult to fully

assess whether such misfits are related to stochasticity associated with small sample sizes (Appendix A) or to inadequacy of the experimental design or to underlying flaws in the formulation leading to the generalized Pareto distribution. Likely all of these are involved. Despite the conceptual simplicity of particles moving bumpety-bump down a rough inclined surface, this is a hard problem.

  We reemphasize that the work reported here is aimed at a probabilistic description of expected particle travel distances. This

is a part of a larger effort to understand and inform the essence of the ingredients of nonlocal formulations of transport. We are not suggesting that the results presented here can be immediately cast as a nonlocal formulation of transport. But in order to progress beyond current formulations, the probabilistic physics of particle motions merits closer examination. For example, this level of understanding provides the basis for justifying a Taylor expansion of the convolution (Furbish and Haff, 2010) to form a local Fokker-Planck-like description of transport assuming an exponential-like distribution of travel distances — with

clarity regarding the limitations of this description. Furthermore, we have focused on the energetics of particles in motion. But this is one of two ingredients of nonlocal formulations. The other involves the probabilistic physics and energetics of particle entrainment — a particularly difficult ingredient to constrain because of the difficulty of observing the entrainment process and because we do not yet know how to properly simulate this process. For this we must rely on theory and measurements of tracer particles in ways that have yet to be designed.

We end with a philosophical point. We enjoy eating our favorite tortilla chips, and mostly we enjoy them with a well prepared dip, for example, spicy guacamole. But let us be honest. The experience then is no longer about the chips, it's about the dip. The chips are just the guacamole delivery system. (Yumm.) Similarly, these companion papers nominally concern particle motions on inclined rough surfaces. But these particles are just the delivery system. The dip consists of the coherent statistical mechanics framework for describing the particle motions, and a demonstration that such a framework, albeit with rough edges,

is possible. This represents a solid basis for subsequent efforts aimed at replication, falsification and refinement or replacement, and possibly for fresh ideas concerning particle motions more generally.

*Code and data availability.* Emmanuel Gabet provided the data described in Section 3.1. The data described in Section 4.1 and Section 4.2 are available from Dibiase et al. (2017) and Roth et al. (2020). The data described in Section 3.3, including video and audio files, and The MATLAB/GNU Octave code described in Appendix A are archived and readily accessible via the Vanderbilt University Institutional

5 Repository (https://ir.vanderbilt.edu/handle/1803/9742).



## Appendix A: Parameter estimation

Here we demonstrate the basis for using visual fits of the exceedance probability plots to illustrate the behavior of the generalized Pareto distribution, and we provide context for interpreting these fits. We work with the dimensionless form of the distribution for comparison with Figure 2, and pursue a straightforward Monte Carlo analysis.

First, let $\hat{x}_u$ denote a random number drawn from a uniform distribution with support $[0, 1]$ and cumulative distribution function $F_{\hat{x}_u}(\hat{x}_u) = \hat{x}_u$. In turn, the cumulative distribution function of the generalized Pareto distribution is

$$F_{\hat{x}}(\hat{x}) = 1 - \frac{b^{1/a}}{(a\hat{x} + b)^{1/a}}. \tag{A1}$$

Equating $F_{\hat{x}_u}(\hat{x}_u)$ and Eq. (A1) leads to

$$\hat{x} = \frac{b}{a}\left[\frac{1}{(1 - \hat{x}_u)^a} - 1\right], \tag{A2}$$

which provides an algorithm for generating values of $\hat{x}$ drawn from the generalized Pareto distribution with shape and scale parameters $a$ and $b$, starting with values of $\hat{x}_u$ selected by a uniform random number generator.

Second, consider a maximum likelihood estimation (MLE) of the values of $a$ and $b$, noting that an MLE is the same as a Bayesian estimate assuming a uniform (maximum entropy) prior distribution of the parametric values. The MLE method is a popular, standard choice for estimating parametric values of distributions, notably heavy-tailed distributions, because of its asymptotic properties of consistency and efficiency. (However, see the delightful review by Cam (1990) regarding maximum likelihood estimates, in particular his nine principles on p. 165.) The shape and scale parameters $a$ and $b$ are not orthogonal. The MLE of $a$ is $\tilde{a} = 1/\tilde{a}_L$ where

$$\tilde{a}_L = \frac{N}{\sum_{i=1}^{N} \ln(1 + \hat{x}_i/\tilde{b}_L)}, \tag{A3}$$

and the MLE estimate of $b$ is $\tilde{b} = \tilde{b}_L/\tilde{a}_L$ where $\tilde{b}_L$ is obtained from an iterative solution of

$$\frac{N}{\tilde{b}_L \sum_{i=1}^{N} \hat{x}_i/(\tilde{b}_L^2 + \tilde{b}_L \hat{x}_i)}$$

$$-\frac{N}{\sum_{i=1}^{N} \ln(1 + \hat{x}_i/\tilde{b}_L)} - 1 = 0. \tag{A4}$$

These are biased estimates (Giles et al., 2013), but they nonetheless provide useful information concerning parameter estimation. This bias increases with decreasing sample size and with increasing censorship of the distribution tail. Moreover, the MLE may not converge near $a = 0$ nor if the sample has a coefficient of variation less than one. For reference below, whereas the Lomax distribution requires that $a_L > 0$, the MLE given by Eq. (A3) may be negative, and therefore provides an estimate of $a < 0$ for the generalized Pareto distribution if $b$ is known. Note also that in the limit of $a \to 0$ the Pareto distribution is replaced with the exponential distribution with mean $\mu_{\hat{x}} = b$. The MLE of the mean $\mu_{\hat{x}}$ of an exponential distribution is just





the sample average,

$$\tilde{b} = \frac{1}{N} \sum_{i=1}^{N} \hat{x}_i \,, \tag{A5}$$

which is an unbiased estimate.

Now consider a sample size of $N = 100$, consistent with the data sets of Gabet and Mendoza (2012) and Roth et al. (2020). We draw 10,000 samples then calculate and plot exceedance probabilities for varying values of the shape parameter $a$, holding

the scale parameter $b$ fixed for convenience. We also calculate the MLE of $a$ using Eq. (A3) for each sample. The MATLAB/GNU Octave code for doing this is available in Supplementary Materials (Vanderbilt University Institutional Repository, https://ir.vanderbilt.edu/handle/1803/9742), and includes an animation of the results.

Plots of estimated exceedance probabilities $R_{\hat{x}}(\hat{x})$ for all samples provide a visual sense of the variability in these probabilities associated with the inherent randomness in drawing samples of $\hat{x}$ from the known distribution (Figure A1). The animation

mentioned above shows the outcome of successive draws, and nicely illustrates that many draws, by chance, bear little resemblance to the theoretically expected exceedance probability function as well as, in particular, the squirrelly behavior of values in the tails. (We hope that the animation drives home the point to avoid over-fitting and over-interpreting data in the tail of a heavy-tailed distribution with small sample size $N$.) Here are key items to consider. First, the variability in calculated exceedance probabilities increases with $a$, that is, with increasing heaviness of the distribution tail. This is not surprising, as

a finite sample size represents a decreasing proportion of the total probability in the distribution as $a$ increases. Second, the variability in the MLE of $a$ increases with increasing $a$ (Figure A2), reflecting the first point above. Aside from the bias of these estimates, the difference between estimates of $a$ and the true value can be large, although proportional differences are similar across values of $a$. For example, with $a = -0.5$, about 32% of the estimated values exceed a difference of $\pm 10\%$ from the true value; with $a = 0$, about 32% exceed a difference of $\pm 10\%$ about the true value of $b = 1$; and with $a = 1$, about 31% of the

estimated values exceed a $\pm 10\%$ difference. Third, the variability in the exceedance probability plots and the MLEs decreases — that is, these values converge to the true values — only when $N$ approaches 10,000 or more (Figure A3). Fourth, with increasing censorship of the data, the MLEs based on the uncensored values become strongly biased (Figure A4). Moreover, this simple demonstration of the inherent variability in estimates of $a$ does not involve the collinear effects and added variability associated with simultaneously estimating $b$. Fifth, despite the variability in the exceedance probability plots, the sign of the

concavity of the plots for large positive or negative $a$ is clear. However, near isothermal conditions ($a = 0$), individual samples could appear to represent net particle heating when in actuality conditions of net cooling exist, and vice versa. Note that in the example of $a \to 0$ (Figure A1, Figure A2), we calculate sample exceedance probabilities and $\tilde{b}$ for the exponential distribution. According to the central limit theorem, values of $\tilde{b}$ are approximately normally distributed with variance $\sim \sigma_{\hat{x}}^2 / N$.

Here is an important sidebar. In the presence of an exact theory that predicts the values of $a$ and $b$, one can appeal to, for example, the central limit theorem or a Monte Carlo analysis (as above) or MLE methods or bootstrapping to assign so-

called confidence estimates associated with these known values of $a$ and $b$. These specifically give information regarding the likelihood that a sample of size $N$ will yield values of $a$ and $b$ that differ from the true values. In contrast, in the absence of an exact theory and a priori knowledge of the true values of $a$ and $b$, one can construct similar confidence estimates based on





values of $a$ and $b$ estimated from a single sample. These specifically give information regarding the likelihood that a second sample of size $N$ will yield values of $a$ and $b$ that differ from those estimated from the first sample. But no method — besides

making $N \rightarrow \infty$ (Cam, 1990) — can provide information regarding how close the first set of estimated values (or the second set) is to the true unknown values. Alas, the literature is awash with confidence estimates, based on a single sample, incorrectly interpreted as measures of likelihood of containing the unknown values (Amrhein et al., 2019).

We therefore reemphasize our objective. At this stage of our work we are aimed at reasonable estimates of the shape and scale parameters in order to demonstrate the existence of the behaviors — rapid thermal collapse, isothermal conditions, and

net heating of particles — represented by the generalized Pareto distribution. Refined values of these parameters are not needed until we possess a clearer understanding of the mechanics of deposition. Semi-log plots highlight deviations in the tails, and provide a clear sense of the concavity that discriminates between cooling and heating. Log-log plots highlight deviations near the origin, and provide a sense of the log-linear decay of the tails for heavy-tailed conditions. The variability in the tails of the distribution as outlined above emphasizes the importance of avoiding over-fitting of the tails in visual fitting (or in any other

method of fitting).

For comparison with our fits, we return to dimensional quantities and compute the MLE values of $A$ and $B$ (Table A1). The MLE is implemented in the "flomax" algorithm in the Renewal Method for Extreme Values Extrapolation library of the R Project for Statistical Computing; it is implemented in the "gpfit" algorithm of the MATLAB programming language; or it can be coded directly from Eq. (A3) and Eq. (A4). With reference to Table A1, the MLE algorithm converges in all cases

using the MATLAB "gpfit" algorithm (but not the R "flomax" algorithm). However, it returns poor (sometimes nonsensical) estimates for $A \lesssim -1/2$ or near $A \approx 0$. The MLE degrades with increasing $A$ and increasing censorship. Also note that the MLE estimates do not necessarily improve the fits (Figure A5), likely due to the relatively small sample sizes and the likelihood that the data represent samples that are strongly censored, that is, where a significant proportion of the distribution is contained in that part of the tail that is not sampled.

One alternatively can choose, say, a nonlinear least-squares fitting algorithm that weights various parts of data differently, emphasizing or deemphasizing values near the origin or in the tails. We suggest, however, that this is just a rule-based version of visual fitting. We also note that visual fitting is not directly influenced by censorship, although the form of the censored tail can never be known (Ballio et al., 2019). Bringing more sophisticated techniques to bear (e.g., Hosking and Wallis, 1987; Castillo and Hadi, 1997; Cramer and Schmiedt, 2011; Giles et al., 2013, 2016; Pak and Mahmoudi, 2018) to refine estimates of

$A$ and $B$ is premature. There is a need to collect larger data sets, avoiding censorship if possible, and only then aim at refined estimates of the parametric values as their theoretical basis is improved. Moreover, any real data set is not immune from the possibility, by chance, of representing a misfit from the underlying distribution and yielding parametric estimates that markedly differ from this underlying distribution — just as with the numerical examples above. But no formal quantitative analysis can reveal or fix this misfit.

5    We end with a cautionary note: Parametric values of a heavy-tailed distribution estimated from a data set with $N < 1000$ (and possible with $N < 10,000$), if presented as being precise, merit a healthy skepticism, particularly if the tail of the distribution is censored. Korup et al. (2012) address a related point in demonstrating how the exponents in power functions involved in





**Table A1.** Fitted and estimated values of the parameters for the data reported by Gabet and Mendoza (2012), the Vanderbilt experiments, DiBiase et al. (2017) and Roth et al. (2020).

| Site | Slope (deg) | Particle size (m) | $A^1$ | $B^1$ (m) | $A^2$ | $B^2$ (m) |
|---|---|---|---|---|---|---|
| Gabet and Mendoza (2012) | 0 | 0.01 | -0.48 | 0.42 | -0.24 | 0.35 |
| | 3 | 0.01 | -0.55 | 0.32 | -0.54* | 0.31* |
| | 6 | 0.01 | -0.36 | 0.30 | -0.32 | 0.28 |
| | 9 | 0.01 | -0.65 | 0.43 | -0.43 | 0.37 |
| | 12 | 0.01 | 0.03 | 0.31 | -0.26 | 0.35 |
| | 15 | 0.01 | 0.02 | 0.39 | -0.20 | 0.46 |
| | 18 | 0.01 | 0.09 | 0.99 | -0.48 | 1.27 |
| | 21 | 0.01 | 0.70 | 2.6 | -1.02* | 2.36* |
| | 24 | 0.01 | 3.6 | 4.7 | -0.71* | 1.76* |
| Vanderbilt | 0 | $A^3$ | -0.54 | 0.033 | -0.27* | 0.027* |
| | 0 | R | -0.51 | 0.049 | -0.52* | 0.051* |
| | 0 | S | -0.49 | 0.035 | -0.51* | 0.039* |
| | 5.1 | A | -0.39 | 0.075 | -0.40 | 0.085 |
| | 5.1 | R | -0.24 | 0.119 | -0.32 | 0.13 |
| | 5.1 | S | -0.51 | 0.098 | -0.50 | 0.10 |
| | 8.5 | A | -0.36 | 0.12 | -0.35 | 0.12 |
| | 8.5 | R | 0.10 | 0.19 | -0.13 | 0.24 |
| | 8.5 | S | -0.35 | 0.12 | -0.38 | 0.13 |
| | 10.2 | A | -0.56 | 0.22 | -0.34 | 0.20 |
| | 10.2 | R | 0.020 | 0.28 | -0.10 | 0.33 |
| | 10.2 | S | -0.51 | 0.24 | -0.45* | 0.23* |
| | 14.0 | A | 0.30 | 0.35 | -0.20 | 0.48 |
| | 14.0 | S | 0.30 | 0.41 | -0.20 | 0.51 |
| | 15.6 | A | 0.77 | 1.18 | -0.65* | 0.99* |

scaling relations may be particularly sensitive to the presence or absence of extreme values in the data sets used to estimate the exponents. To quote the sixth of nine delightful principles offered by Cam (1990), "Never trust an estimate which is thrown out of whack if you suppress a single observation." Stumpf and Porter (2012) suggest that more generally statistically fitted power laws have little more than anecdotal value in the absence of a theoretical basis.





**Table A2.** This is actually not a new table, but rather a continuation of **Table A1**, separated here for the single-column format. The plan is to combine these tables together as one for the double-column format.

| | | | | | |
|---|---|---|---|---|---|
| DiBiase et al. (2017) | 38 | 0.025 | 0.81 | 2.4 | 0.159 | 2.82 |
| | 38 | 0.05 | 1.7 | 5.1 | -0.41* | 5.9* |
| | 38 | 0.10 | 5.0 | 8.8 | -1.19* | 9.5* |
| Roth et al. (2020) V | 0 | all | -0.41 | 0.087 | -0.64* | 0.16* |
| | 14 | 0.017 | -0.41 | 0.165 | -0.18 | 0.14 |
| | 14 | 0.045 | 0.45 | 0.23 | 0.32 | 0.23 |
| | 14 | 0.073 | 1.1 | 0.13 | 5.82 | 0.00017 |
| | 20 | 0.017 | -0.23 | 0.72 | -0.20 | 0.73 |
| | 20 | 0.045 | -0.30 | 1.8 | -0.93* | 3.0* |
| | 20 | 0.073 | 0.20 | 1.0 | -1.11* | 3.4* |
| | 24 | 0.017 | -0.06 | 0.60 | -0.17 | 0.68 |
| | 24 | 0.045 | -0.01 | 2.3 | -0.16 | 2.7 |
| | 24 | 0.073 | 0.01 | 3.4 | -0.22 | 4.1 |
| | 39 | 0.045 | -0.12 | 0.30 | -0.056 | 0.32 |
| | 39 | 0.017 | -0.38 | 3.7 | -0.17 | 3.2 |
| | 39 | 0.073 | 0.70 | 4.8 | -1.12* | 18.7* |
| Roth et al. (2020) B | 17 | 0.017 | -0.39 | 0.27 | -0.39 | 0.27 |
| | 17 | 0.045 | -0.03 | 0.49 | 0.36 | 0.45 |
| | 17 | 0.073 | 0.67 | 0.39 | 0.59 | 0.47 |
| | 20 | 0.017 | 0.10 | 0.18 | 0.18 | 0.17 |
| | 20 | 0.045 | 1.30 | 0.90 | 1.19 | 0.89 |
| | 20 | 0.073 | 1.68 | 0.64 | 1.04 | 0.71 |

[1] Estimated visually; values reproduced from Table 00.

[2] Estimated from MLE algorithm; asterisk denotes problematic estimate due to $A \lesssim -0.5$, $A \to 0$ or censored data.

[3] A = large angular, R = large rounded, S = small.

## 5  Appendix B:  Particle launching conditions

Consider as an example the initial exceedance probability plot for the angular particles on a flat surface (Figure B1a), which shows a clear inflection at about 5 cm. In this example, high-speed imaging of particles launched from the catapult reveal that the particles consistently travel $\sim 2$ cm horizontally before their first collisions with the surface, as expected from calculations using Newton's second law for measured initial velocities. (Free-flight distances increase with increasing surface slope.) These initial flights involve negligible rotational motion. The particles then experience widely varying changes in their motions over the next 2–3 cm following the first collisions, often with the onset of rotational motion. In the text we suggest that the inflection in the exceedance probability plot reflects the uniformity of the launch velocities followed by a finite distance over which





randomization of the motions occurs. This is consistent with the idea that the factor $\gamma \sim 1$ before randomization occurs, giving an initial disentrainment rate that is smaller than after randomization. However, we note that the inflection also may involve other effects.

For these reasons we truncate the plots at the inflection position, then recalculate exceedance probabilities with reduced $N$ (Figure B1b). In this example the truncation distance is a significant proportion of the total travel distances. However, this
truncation distance is less important when effects of initial (near-launch) conditions occur over a distance that is small relative to total travel distances. Unfolding the details of the physics of particle motions over short distances is for a later time.

## Appendix C: Uncertainty in calculated quantities

Of interest is how uncertainty in the estimates of the shape and scale parameters, $A$ and $B$, propagates to uncertainty in the calculated values of $\mu$, $\alpha$, $Ki$ and $Ki_*$. Because $A$ and $B$ are obtained by visual fitting, for illustration we conservatively assume
that the standard deviations of these values associated with a great number of samples of similar size $N$ vary as $A/\sqrt{N}$ and $B/\sqrt{N}$ based on Monte Carlo simulations as described in Appendix A. This effectively assumes the coefficient of variation formed by the sampling standard deviation is $\sim 1/\sqrt{N}$. This provides the basis for gaining a sense of the relative magnitude of the variability in the calculations of $\mu$, $\alpha$, $Ki$ and $Ki_*$. For the Vanderbilt data (Section 3.3) we also incorporate the uncertainty in the launch velocities $u_0$ provided in Table 5.

We perform a straightforward Mont Carlo analysis. Assuming values of $A$, $B$ and $u_0$ are approximately Gaussian, we successively solve Eq. (32), Eq. (33), Eq. (8) and Eq. (23) 10,000 times, then calculate the associated coefficients of variation of $\mu$, $\alpha$, $Ki$ and $Ki_*$. Because $A$ and $B$ are obtained by visual fitting (as opposed to being based on MLE values) where the number of censored data are included in calculations of exceedance probabilities, we include censored data in setting $N$.

We emphasize that these calculations provide a sense of the variability *only* associated with that of $A$ and $B$ as this cascades
through the successive calculations of $\mu$, $\alpha$, $Ki$ and $Ki_*$. For example, based on Eq. (32) the variability in $\mu$ reflects that in both $A$ and $B$. Based on Eq. (33), the variability in $\alpha$ reflects that in $B$ and the variability in $\mu$ previously calculated. Also note that, starting with Eq. (32), as the slope $S$ increases the relative contribution of this fixed term to calculated values of $\mu$ increases. These calculations do *not* represent the natural variability in $\mu$, $\alpha$, $Ki$ and $Ki_*$ if these quantities somehow could be measured directly, independently of $A$ and $B$.

In general, calculated coefficients of variation decrease with increasing surface slope. Coefficients of variation are on the order of 10% or more for smaller slopes in the experiments reported by Gabet and Mendoza (2012) and in the Vanderbilt experiments. Coefficients of variation generally are on the order of 1% or smaller in the field-based experiments of DeBiase et al. (2017) and Roth et al. (2020) involving steep slopes.





## Appendix D: Particle energy balance

Let $E_{p0} = mgh = (1/2)mw_0^2$ denote the initial impact energy of a particle with mass $m$ falling from height $h$ onto a horizontal
surface, where $w_0$ is the vertical impact velocity. Then let $w_1$ and $u_1$ denote the vertical and horizontal rebound velocity
components. Assuming negligible rotational energy during the initial free fall, the energy balance may be written as

$$E_{p0} = f_c + \frac{1}{2}mu_1^2 + \frac{1}{2}mw_1^2 + \frac{1}{2}I\omega^2 \,, \tag{D1}$$

where $f_c$ is the frictional loss due to particle-surface deformation, $I$ is the moment of inertia and $\omega$ is the angular velocity. If
we set the initial and final vertical positions of the rebounding motion at $z(0) = z(T) = 0$, then from Newton's second law,

$$w_1 = \frac{g}{2}T \,, \tag{D2}$$

where $T$ is the travel time to the second collision. We assume as an approximation that $f_c = (1 - \epsilon^2)E_{p0}$, where $\epsilon$ is the normal
coefficient of restitution. This effectively assumes that the energy loss due to particle-surface deformation is the same as that
of a collinear collision. Now Eq. (D1) becomes

$$\epsilon^2 E_{p0} = \frac{1}{8}mg^2 T^2 + \frac{1}{2}mu_1^2 + \frac{1}{2}I\omega^2 \,. \tag{D3}$$

We can experimentally determine $\epsilon$, $E_{p0}$ and $T$ for individual particles. However, we generally cannot determine $u_1$ from side
imaging (except by chance with motion transverse to the camera). We also cannot readily determine $I$ for irregular particles,
nor $\omega$ from side imaging. Solving Eq. (D3) for the last two terms then represents the conversion $E_c$ of translational kinetic
energy just prior to impact into translational energy associated with surface parallel motion and rotational energy. That is,

$$E_c = \epsilon^2 E_{p0} - \frac{1}{8}mg^2 T^2 \,. \tag{D4}$$

## Appendix E: Effect of launch velocity

For completeness, here we show plots of the friction factor $\mu$ versus the slope $S$ and the factor $\alpha$ versus the Kirkby number $Ki$
using the initial launch velocity $u_0$ rather than a reduced velocity in the calculations as presented in Figure 17 and Figure 18.

Values of $\mu$, notable at smaller slopes $S$ (Figure E1), are noticeably larger than those calculated in Figure 17. The values
appear to converge to the 1:1 line with increasing slope as $Ki \to 1$, as in Figure 17.

Values of $\alpha$ and associated Kirkby numbers tend to be smaller (Figure E2) than those calculated in Figure 18. However, the
overall variation between $\alpha$ and $Ki$ is similar.

## Appendix F: The Pareto distribution as a mixture of exponential distributions

It is well known that a Pareto distribution with positive shape parameter can be obtained as a mixture of exponential distribu-
tions whose rate parameters are distributed as a gamma distribution. This result suggests an interesting physical interpretation of





the Pareto distribution of particle travel distances, and it also may indicate a strategy for clarifying how particle size and shape in concert with surface roughness influence the extraction of particle energy and the likelihood of deposition. For completeness we therefore offer the following.

Recall that for an exponential distribution of travel distances $x$ the fixed disentrainment rate is $P_x = 1/\mu_x$. We then write the conditional distribution as

$$f_{x|P_x}(x|P_x) = P_x e^{-P_x x} \,. \tag{F1}$$

We may now treat the rate $P_x$ as a random variable that is distributed as a gamma distribution, namely,

$$f_{P_x}(P_x; A_g, B_g) = \frac{B_g^{A_g}}{\Gamma(A_g)} P_x^{A_g - 1} e^{-B_g P_x} \,, \tag{F2}$$

with shape parameter $A_g$ and scale parameter $B_g$. The unconditional distribution of travel distances $x$ is then obtained as a gamma weighting of the conditional exponential distribution, namely,

$$f_x(x) = \int_0^\infty f_{x|P_x}(x|P_x) f_{P_x}(P_x; A_g, B_g) \, \mathrm{d}P_x \,. \tag{F3}$$

Substituting Eq. (F1) and Eq. (F2) into Eq. (F3) and evaluating the integral then leads to

$$f_x(x) = \frac{A_g B_g^{A_g}}{(x + B_g)^{1 + A_g}} \,. \tag{F4}$$

This is a Lomax distribution (compare with Eq. (28)) with shape parameter $A_g = 1/A$ and scale parameter $B_g = B/A = BA_g$.

The expected value $\mu_{P_x} = A_g/B_g = 1/B$ and the variance is $\sigma_{P_x}^2 = A_g/B_g^2 = A/B^2$. This immediately implies that an experimental estimate of $B$ provides an estimate of the expected disentrainment rate $\mathrm{E}(P_x) = \mu_{P_x}$; and an estimate of $A$ together with $B$ provides an estimate of the variance of $P_x$.

Because $P_x$ is a random variable, and because the exponential distribution, Eq. (F1), implies isothermal conditions, we now
use Eq. (16) to write

$$P_x = \frac{\gamma m g \mu \cos\theta}{\alpha E_{a0}} \tag{F5}$$

which is the reciprocal of the deposition length $L_c$ with specified average energy $E_{a0}$. For a given particle mass $m$, slope angle $\theta$ and energy $E_{a_0}$, the quantities $\gamma$, $\mu$ and $\alpha$ are random variables. That is, we may envision an ensemble of combinations of these quantities, each of which yields isothermal conditions. In turn, envision a great number (cohort) of particles. Then
$f_{P_x}(P_x; \alpha, \beta) \mathrm{d}P_x$ is the probability that particles possess the value $P_x$. These particles are deposited exponentially with mean $\mu_x = 1/P_x$. When combined with all other exponential distributions with varying means (i.e., different combinations of $\gamma$, $\mu$ and $\alpha$), the collective effect is a Pareto distribution. Each subset of particles with rate $P_x$ behaves isothermally, but collectively the downslope energy variation of the entire cohort involves net heating.

As an example, for a value of $A = 0.01$ representing nearly isothermal conditions, the gamma distribution of $P_x$ is centered about the value of $1/B$ (Figure E1a) and approaches a Dirac function in the limit of $A \to 0$ as the variance $\sigma_{P_x}^2 = A/B^2 \to 0$.





This represents the exponential limit of a Pareto (or Lomax) distribution. As $A$ increases, the distribution of the disentrainment
rate $P_x$ becomes increasingly skewed toward $P_x = 0$, which collectively gives a heavy-tailed Pareto distribution. In turn, the
distribution of the reciprocal $\mu_x = 1/P_x$ is given by the inverse gamma distribution (Figure E1b). Again, in the limit of $A \to 0$
the mean travel distances $\mu_x$ of the mixture of exponential distributions converge to the mean value of the Pareto distribution,
namely, $B$. With increasing $A$ the mixture of mean values is distributed about the mean, notably incorporating increasingly
larger values of $\mu_x$.

Inasmuch as the quantities $\gamma$, $\mu$ and $\alpha$ can be related to measurable quantities — for example, particle size, particle shape and
surface roughness — then Eq. (F5) suggests the possibility of formulating a multiplicative relation between these properties
and the shape parameter $B = 1/\mathrm{E}(P_x)$. An initial effort to this effect is reported by Roth et al. (2020).

*Author contributions.* All authors contributed to the conceptualization of the problem and its technical elements. SGW led the VU experiments. DJF wrote much of the paper with contributions by the other authors.

*Competing interests.* We have no competing interests.

*Acknowledgements.* We acknowledge support by the U.S. National Science Foundation (EAR-1420831 and EAR-1735992 to DJF, CNS-1831770 to JJR and EAR-1625311 to DLR). We appreciate critical discussions with Angel Abbott (1990–2018) concerning rarefied particle motions on the spectacular scree slopes within Martian craters, with Jonathan Gilligan concerning parameter estimation, and with Mark Schmeeckle concerning collision mechanics. Emmanuel Gabet generously provided the experimental data in Figures 3 and 4. We appreciate
the help of Brandt Gibson in setting up the experimental slope, and the help of Rachel Bain in coding the QQ algorithm.





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



**Figure 11.** Plots of exceedance probability versus travel distance for the Vanderbilt experiments over six values of slope $S$ showing angular (open circles), rounded (black circles) and small (gray circles) particles together with fitted distributions (lines).



Earth **Surface**
**Dynamics**
Discussions



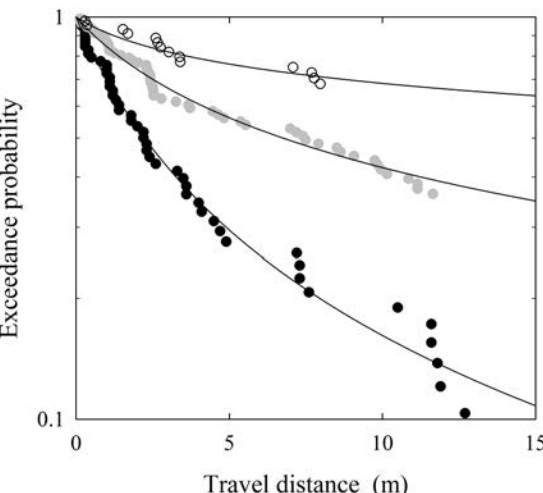

**Figure 12.** Plot of exceedance probability versus travel distance for experiments described by DiBiase et al. (2017) showing small (black circles), medium (gray circles) and large (open circles) particles together with fitted distributions (lines).

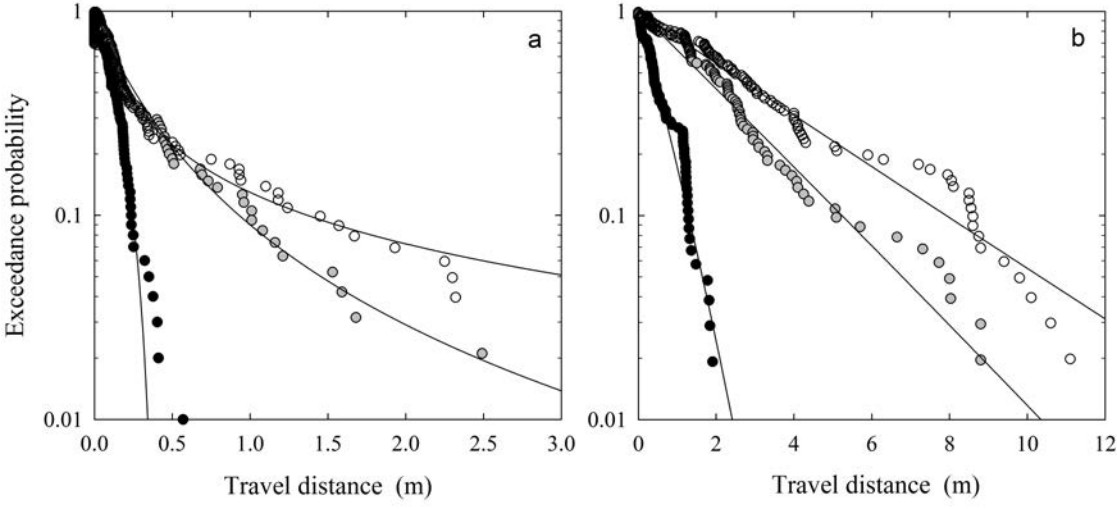

**Figure 13.** Plot of logarithm of exceedance probability $R_x(x)$ versus travel distance $x$ for experiments described by Roth et al. (2020). These examples are for sites V14 (a) and V24 (b) showing data for small (black circles), medium (gray circles) and large (open circles) particle sizes, together with fitted distributions (lines).





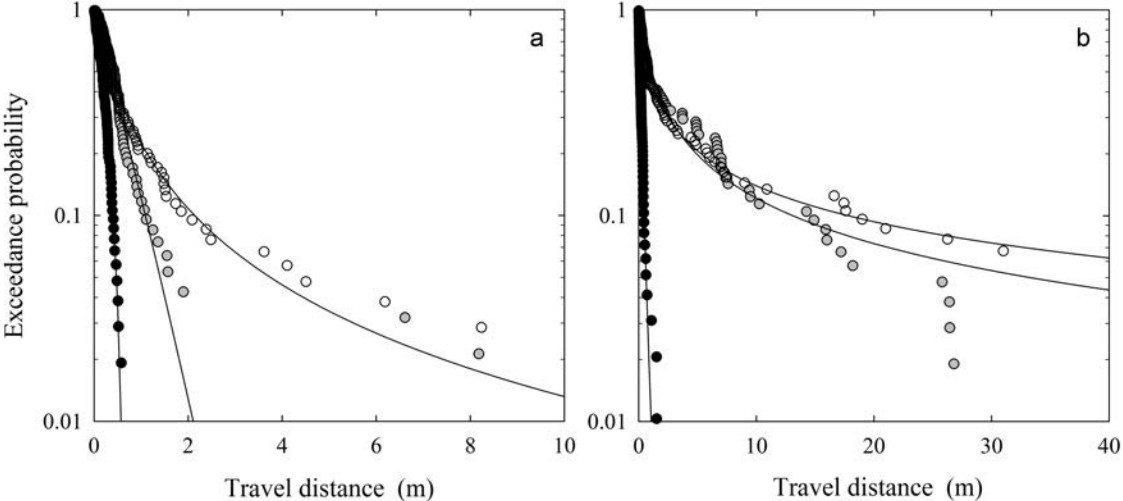

**Figure 14.** Plot of logarithm of exceedance probability $R_x(x)$ versus travel distance $x$ for experiments described by Roth et al. (2020). These examples are for sites B17 (a) and B20 (b) showing data for small (black circles), medium (gray circles) and large (open circles) particle sizes, together with fitted distributions (lines).

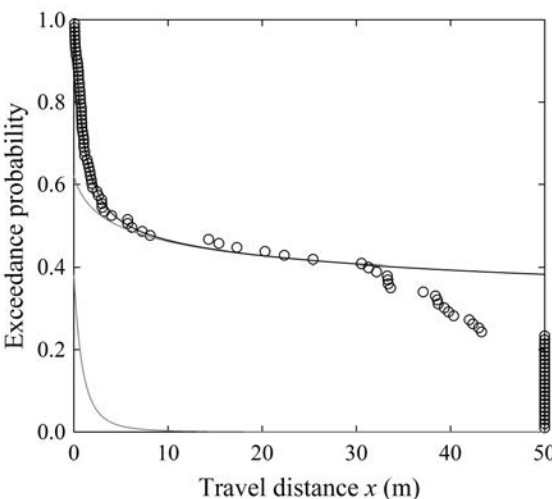

**Figure 15.** Plot of exceedance probability versus travel distance for experiment B28M described by Roth et al. (2020) showing data (circles) fit to mixed distribution (black line) composed of sum of two distributions (gray lines) weighted by proportions $p_1$ and $p_2 = 1 - p_1$. Note effect of increased frictional cooling after slope inflection at $\sim 33$ m; data at $x = 50$ m are censored, but included for reference. Plots for B28S and B28L (Table 9) are similar in appearance.

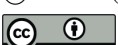



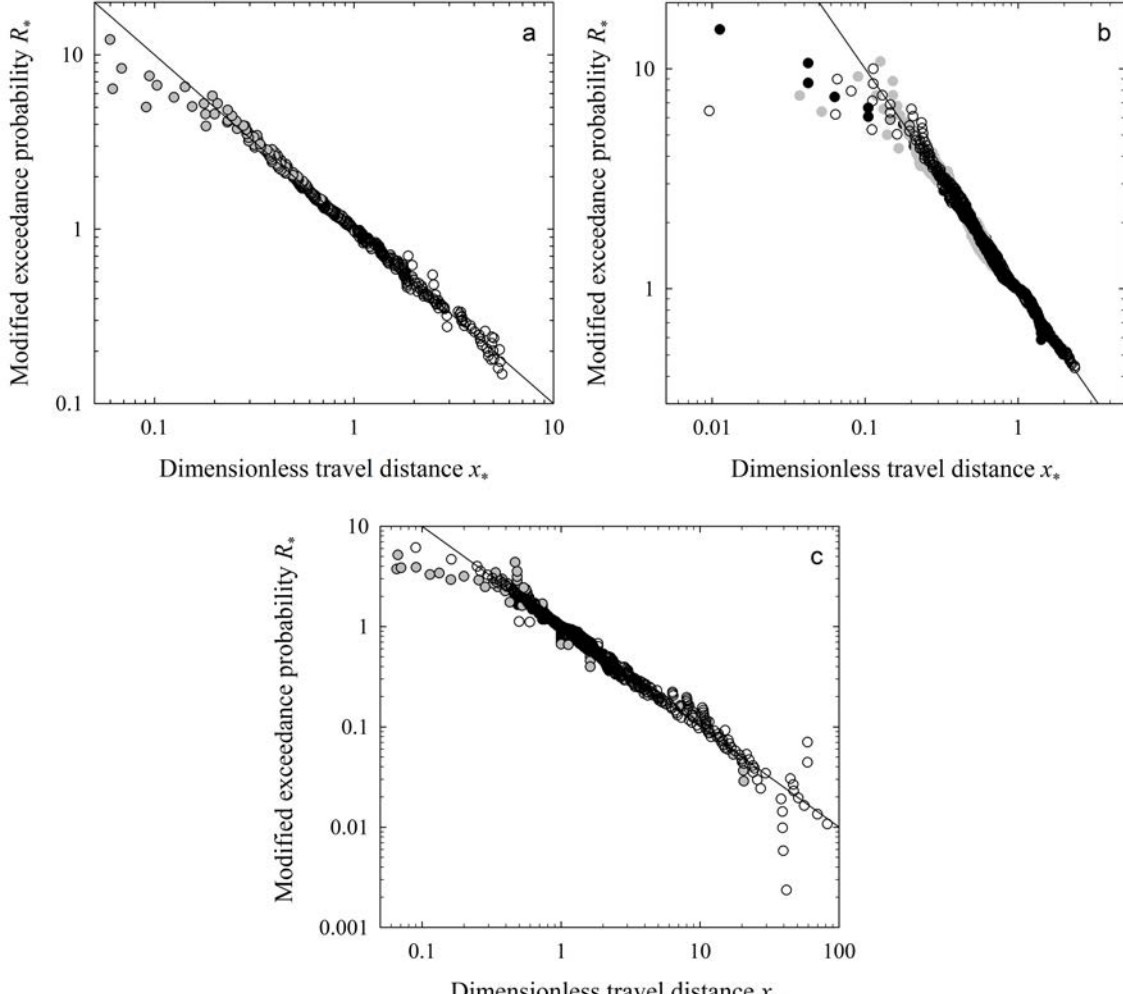

**Figure 16.** Plot of modified exceedance probability $R_*$ versus dimensionless travel distance $x_*$ and line with log-log slope of $-1$ for (a) experiments described by Gabet and Mendoza (2012) (gray circles) and experiments described by DiBiase et al. (2017) (open circles), (b) Vanderbilt experiments, and (c) experiments described by Roth et al. (2020) for V sites (gray circles) and B sites (open circles). Data for $A < 0$ fall to left of $x_* = 1$ with values in the tails represented by smaller values of $x_*$. Data for $A > 0$ fall to the right of $x_* = 1$ with values in the tails represented by larger values of $x_*$. Total data numbers are (a) $N = 813$, (b) $N = 2980$ and (c) $N = 1878$.

Earth **Surface**
**Dynamics**
Discussions

EGU

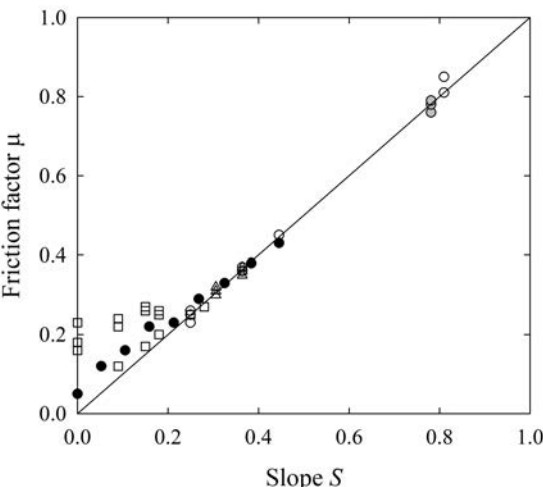

**Figure 17.** Plot of friction factor $\mu$ versus slope $S$ for laboratory experiments described by Gabet and Mendoza (2012) (black circles) and Vanderbilt data (open squares), and field-based experiments of DiBiase et al. (2017) (gray circles) and Roth et al. (2020) for V sites (open circles) and B sites (open triangles) together with 1:1 line.

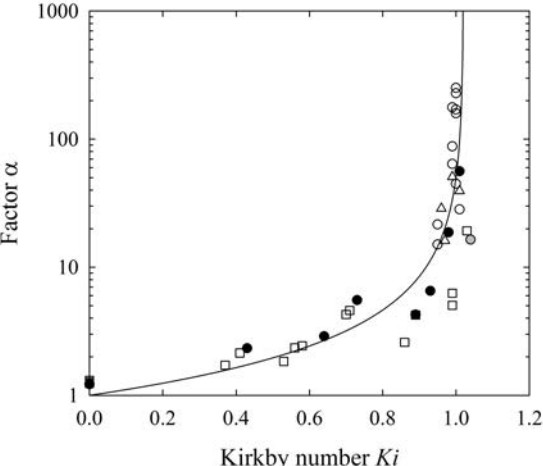

**Figure 18.** Plot of factor $\alpha$ versus Kirkby number $Ki$ for experiments described by Gabet and Mendoza (2012) (black circles), Vanderbilt data (open squares), DiBiase et al. (2017) (gray circles) and Roth et al. (2020) for V sites (open circles) and B sites (open triangles) together with function $\alpha = \alpha_0/(1 - \mu_1 Ki)$ with $\alpha_0 = 1$ and $\mu_1 = 0.98$ (line). Only data for $A < 1$ are included.



Earth **Surface**
**Dynamics**
Discussions



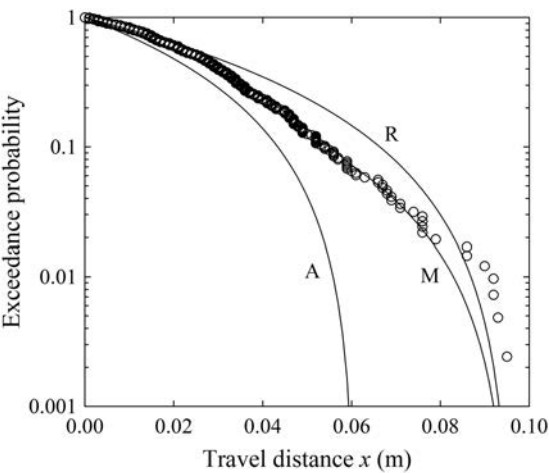

**Figure 19.** Plot of exceedance probability versus travel distance for VU experiment with $S = 0$ showing data (circles) fit to mixed distribution (M) composed of sum of distributions of angular particles (A) and rounded particles (R) depicted in Figure 11.





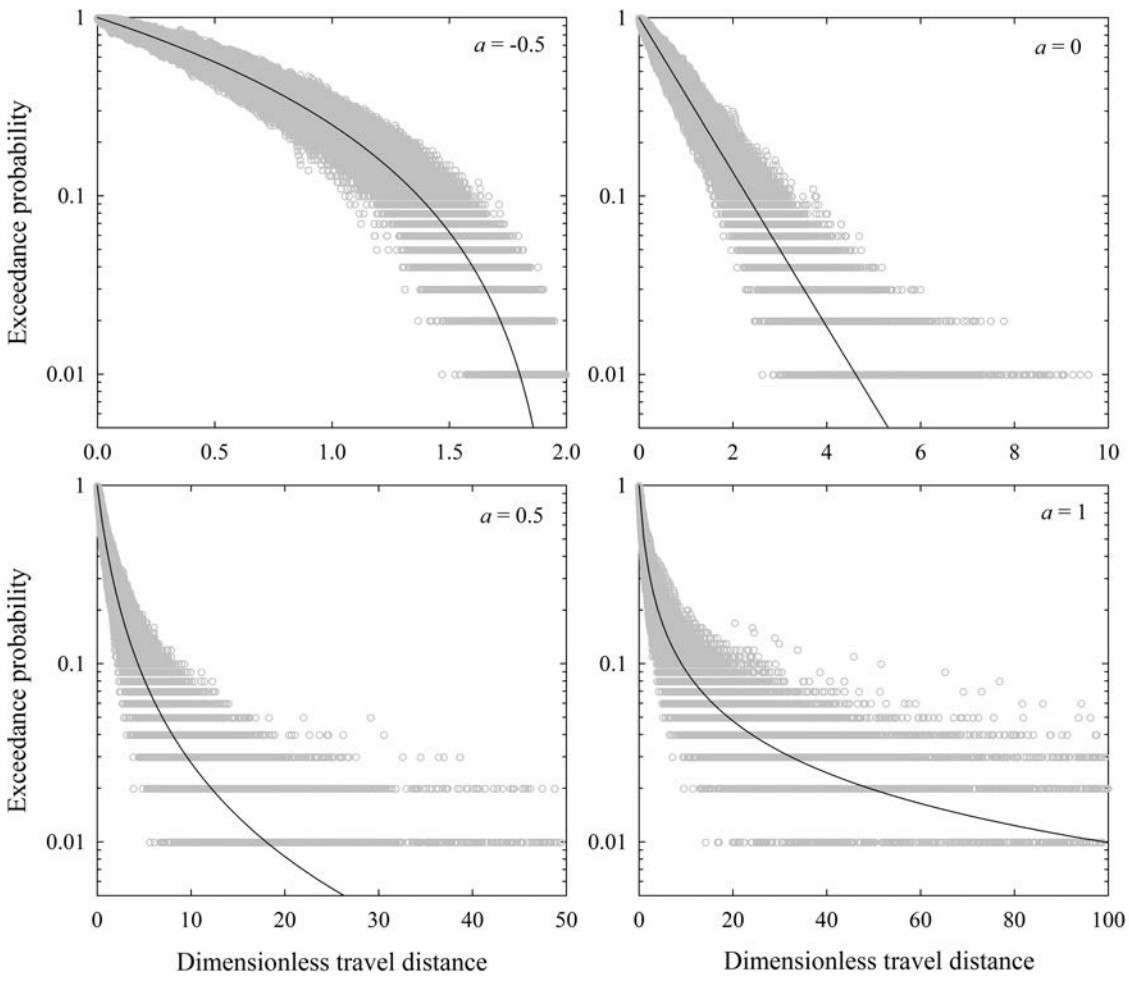

**Figure A1.** Plots of exceedance probability $R_{\hat{x}}(\hat{x})$ versus dimensionless travel distance $\hat{x}$ for different values of the shape parameter $a$ assuming scale parameter $b = 1$. Each plot shows 1,000 samples, each of size $n = 100$, together with theoretical exceedance probability function (line).



Earth **Surface**
**Dynamics**
Discussions

EGU

**Figure A2.** Histograms of maximum likelihood estimates of shape parameter $a$ assuming scale parameter $b = 1$, and maximum likelihood estimate of scale parameter $b$ for $a = 0$. Each histogram is based on 10,000 samples.





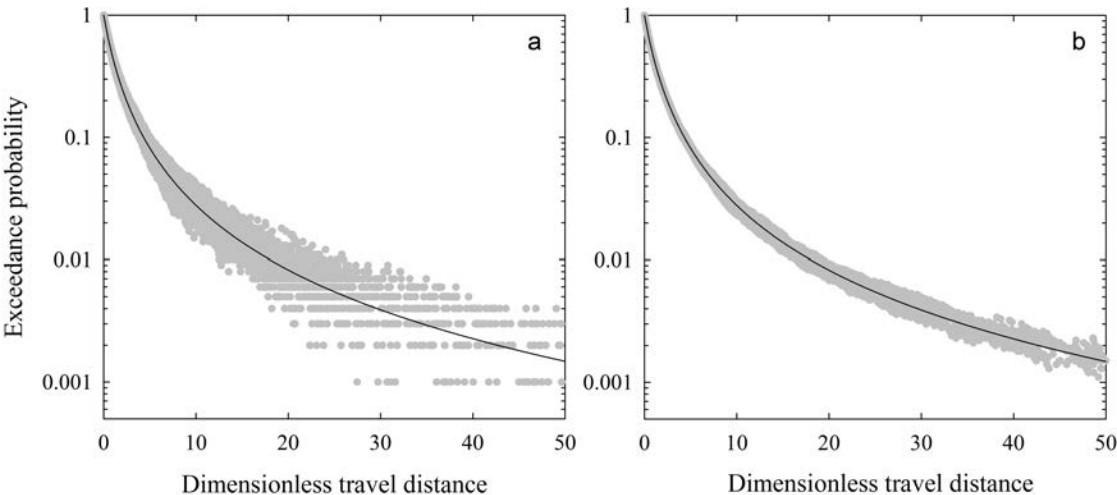

**Figure A3.** Plots of exceedance probability $R_{\hat{x}}(\hat{x})$ versus dimensionless travel distance $\hat{x}$ for shape parameter $a = 0.5$ assuming scale parameter $b = 1$, showing convergence to theoretical exceedance probability function (lines) with increasing sample size $N$. Examples involve (a) 100 samples each of size $N = 1,000$ and (b) 20 samples each of size $N = 10,000$.

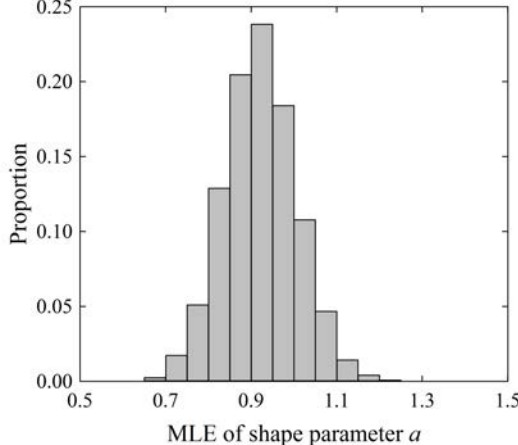

**Figure A4.** Histograms of maximum likelihood estimates of shape parameter $a = 1$ assuming scale parameter $b = 1$ with censorship at $\hat{x} = 50$ based on 10,000 samples. The bias increases as the censorship distance decreases. Compare with Figure A2.





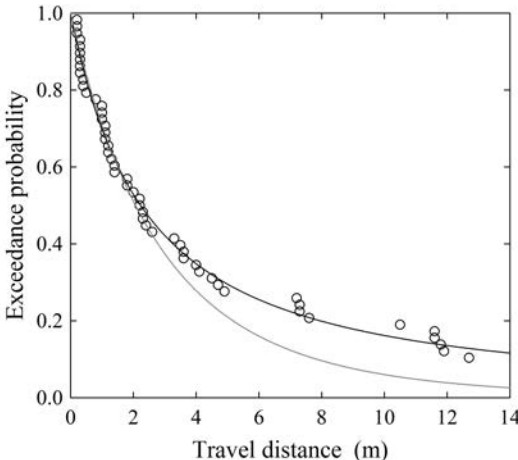

**Figure A5.** Example of fit (gray line) based on MLE values of $A$ and $B$ versus visual fit (black line). This example coincides with the smallest particle size reported by DiBiase et al. (2017) (Table 7).

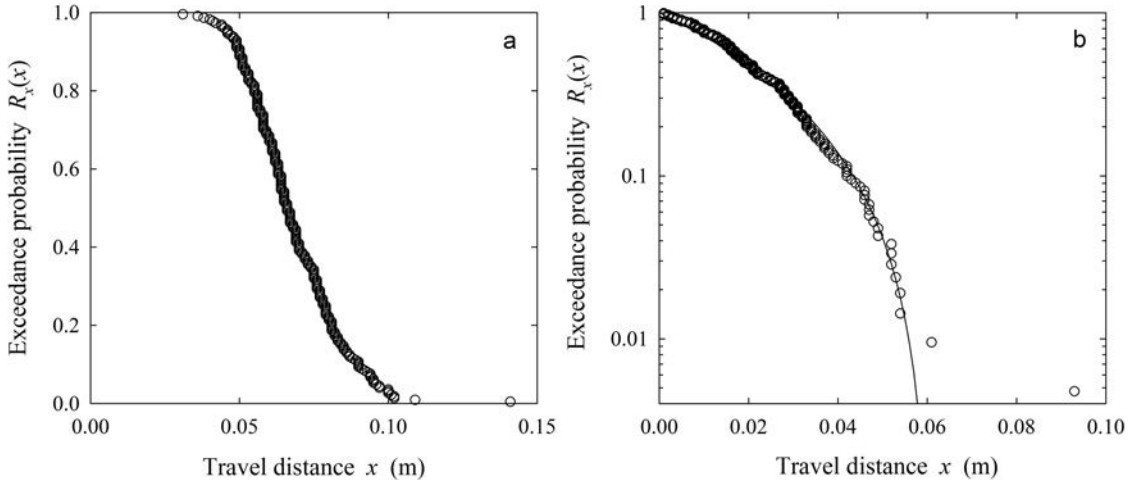

**Figure B1.** Plot of exceedance probability $R_x(x)$ versus travel distance $x$ for the example of angular particles on a flat surface showing (a) initial data set and (b) truncated data set with fitted distribution (line).



Earth **Surface**
Dynamics
Discussions
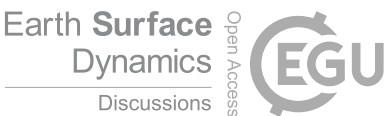

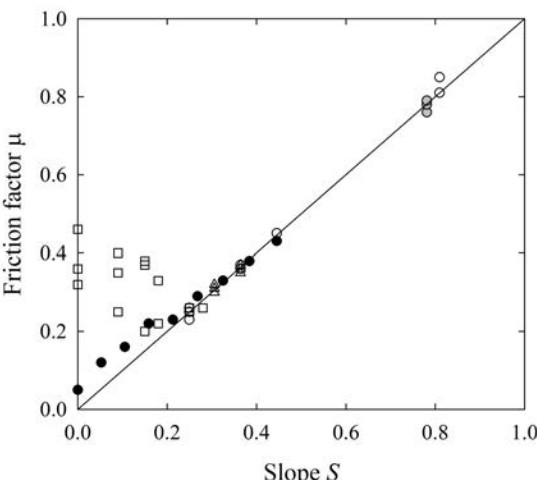

**Figure E1.** Plot of friction factor $\mu$ versus slope $S$ for data shown in Figure 17 using the initial launch velocity $u_0$ in the calculations.

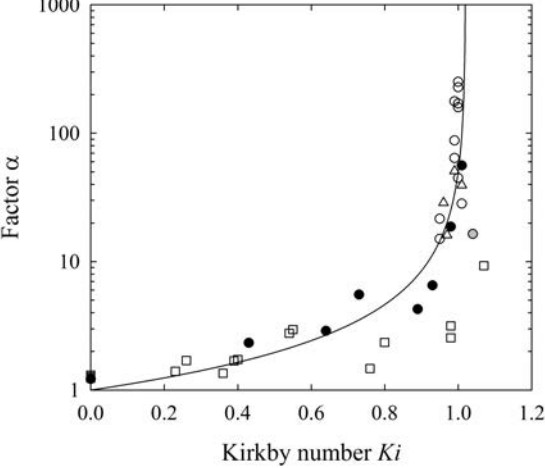

**Figure E2.** Plot of factor $\alpha$ versus Kirkby number $Ki$ for data shown in Figure 18 using the initial launch velocity $u_0$ in the calculations.





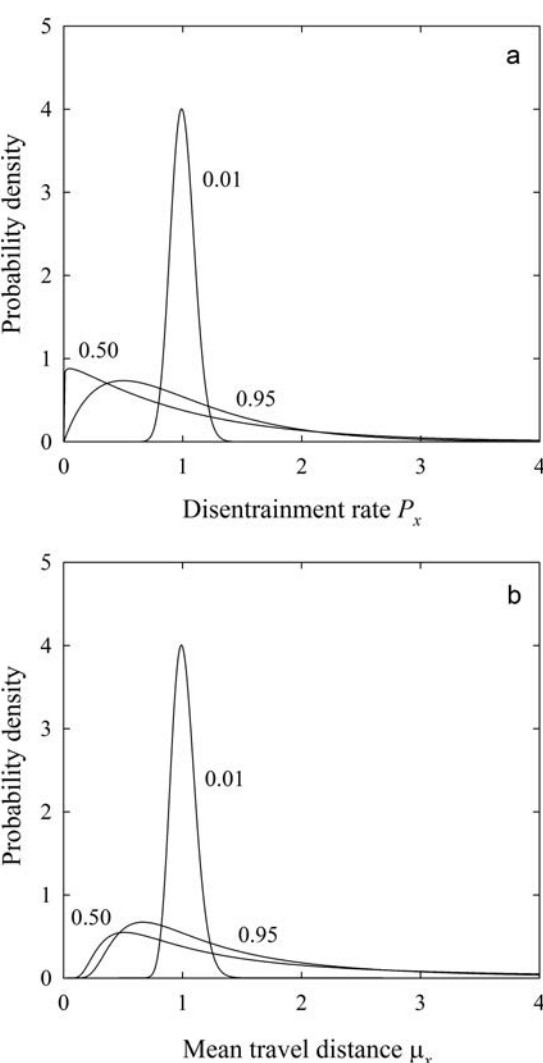

**Figure F1.** Plot of (a) probability density $f_{P_x}(P_x; A_g, B_g)$ of disentrainment rate $P_x$ and (b) probability density of mean travel distance $\mu_x = 1/P_x$ for $A = 0.01, 0.5, 0.95$ $(A_g = 100, 2, 1.05)$ with $B = 1$.