# Peer review of "Rarefied particle motions on hillslopes: 2. Analysis"

_Earth Surface Dynamics, 2020_

## Referee Comment (RC1) · Joris Heyman (Referee) · 1 Feb 2021

2) Analysis :

The second companion paper present results from an experimental study of particle travel distances down a slope, launched by a catapult system. Data is compared to previous experiments and field studies in an exhaustive manner and tested again the theoretical elements provided in the first companion paper (e.g. the expected Pareto distribution of travel distances). Data is well presented and well detailed so that I believe the 2nd study can be published within minor changes.

First, I do not exactly see why high speed imaging is used apart from determining launched velocity. Indeed, all the results shown in Figures present travel distances that

can all be determined without video.

Second, I am not sure to understand how the Pareto fits to the experimental distributions are obtained : by fitting the Pareto parameters, or by estimating them independently with high speed imaging (such as the \beta_z collision restitution parameter) ? I believe the theory would prove very robust if all parameters could be estimated independently via imaging (or other technique). This point is not clear enough and I would suggest the authors to clarify this while presenting their experiments.

Third, it is somewhat disappointing not to see any particle trajectory plotted, that would show the 'heating' (acceleration) for steep slopes, or 'cooling' for milder slopes. I believe much information can be extracted from an acceleration / velocity diagram, as was done for bedload transport in the authors' 2012 paper serie.

Other comments: Fig 9 and 10 (and maybe others) : recall what is \beta_z in the caption so that each figure is understandable by itself.

Please also note the supplement to this comment:
https://esurf.copernicus.org/preprints/esurf-2020-99/esurf-2020-99-RC1-supplement.pdf

**Supplement:**

Review of "Rarefied Particle motions on hillslopes" (Joris Heyman)

Global comments:

These 4[th] companion papers are all very relevant for the different messages and new results they convey. I have fully enjoyed this tough but inspiring reading. I have no major comments to make, although, as explained in part 3 and 4, I suggest submitting the last two studies separately (in esurf or other journal), since their scope is much more general than the hill-slope problem.

While pleasant, the writing style contains many "didactic sidebars", "anecdotes" or humor that do not ease the understanding of an already complex message. I sometimes felt more like reading a book than a journal article (the 4[th] papers format do not help concision neither). Beside precise structural points (see part 1 of the review), I would tend to think that it is possible to globally shorten the text, summarizing the ides, without altering the important results and transferring extra materials in supplementary material.

Specific comments:

*1) Theory*

This first companion paper is the master piece of the serie, presenting all theoretical developments.

**State of the art** The literature review has been placed after the theoretical developments (Section 5 Related formulation), which, in my opinion, do not help to globally envision the originality of the proposed formulation with respect to existing ones, and understand the main challenges of the hillslope problem. I would suggest the authors to better highlight the originality of their approach based on a succinct literature review from the very beginning. This could also help to introduce the important variables.

**Summary of findings** In addition to this originality statement, I believe that a simple summary of findings should precede the detailed theoretical developments. In contrast to the book format, we expect in a journal article to have a rapid understanding of the main results. I had to wait for the summary provided in the second companion paper to make me a clear mental image of the main ingredients of the theory proposed, which I have expressed this way :
1. Particle Mass conservation $dN/dx = - N/E_a$
2. The variation of the ensemble average energy is constant (since forces are constant ?): $dE_a/dx = Cst$
$\rightarrow E_a = Ax+B$
Thus, the mean disentrainment rate is $P = - 1/N \, dN/dx = 1/(Ax+B)$, and the PDF of travel distances is a Pareto distribution, in place of the classical Exponential distribution found when P is a constant. Such ultra-simplified preamble would ease a lot the navigation into the details of the theory latter on.

**Terminology** I understand the analogy between statistical physics of gas and motion of particles down a slope, although I am a bit skeptical on translating all the technical vocabulary for this situation. For instance, the terms *"thermal collapse", "iso-thermal"* and *"net heating"* are not fully transparent with respect to gravity driven motions, and will remain obscure for a majority of readers. In my opinion the notion of "heat" in a gas refers to zero-mean velocity fluctuations, and is thus not perfectly suited to describe a net shift of mean velocities as is the case in non-equilibrium particle motion driven by gravity. I understand the authors conceive the thermal collapse as a net decrease of particle energy and the heating as a net increase of particle energy. However, if they would extend their statistical formulation to the evolution of higher statistical moments of energy states, there will be a confusion

between drift (mean velocity) and diffusion (fluctuations around the mean). My suggestion would be to simply use the transparent terms of mean "deceleration" and "acceleration" of particles ? One of the drawback of using energy balance instead of mass and momentum conservation is that well defined (and measurable) variables such as particle velocity and acceleration are lumped into an energy state, which is less tangible to the observer. Then, it is very easy to understand the disentrainment rate in terms of a decelerating particle (disentrainment probability growing with x, A>0) or accelerating motion (disentrainment probability decreasing with x, A<0).

**Fokker-Planck equation** I understand the authors objective to cast their analysis into a fully probabilistic framework, although I did not get the necessity here to derive a complete Fokker-Planck equation for E if none of the higher moment are used latter on. Indeed, the authors introduce beta^2 (diffusivity of the energy state), which is never used afterwards. Why ? In my opinion, the shape of the pdf (Pareto) is only dependent upon the evolution of the disentrainment rate probability, not on the FP description of energy states. This is a 'simple' non-homogeneous Poisson process. Introducing the FP formulation is thus somewhat confusing for the main message. If this FP equation had an importance for the description of the difference between harmonic or algebraic average of the energy states (Ea, Eh), it might have been preferable to introduce this concept differently (I personally did not get this distinction entirely).

**Meta-stability :** Being familiar to the study of Quartier et al. 2000, I wondered if the theoretical description proposed by the authors is also able to explain the occurrence of meta-stable states of motion due to micro- roughness. Indeed, depending on the initial particle velocity, a particle may be trapped by bed roughness or continue its motion indefinitely. I would have liked to find a mention of this somewhere in the text.

Quartier, L., et al. "Dynamics of a grain on a sandpile model." *Physical Review E* 62.6 (2000): 8299.

Specific Points :
- p5 l8 : I did not get in which sense these probabilistic formulation are "scale independent"
- p5 l17 : "can be a **constant** determined"
- p8 l9 : "The law of the unconscious statistician" ...which means for an unconscious reader ?
- p9 l15: This sidebar could come before, at the beginning of the section
- p10 l20 : "So bear with us" . This do not presage good...
- p11 l25 : Think of moving this didactic sidebar in annexe
- p 12 l 23 : What does "immaterial" mean in this context ?
-p15 : The authors mention "deposition" in granular gases. I do not understand well how particles can deposit in absence of boundary. Do the authors mean "aggregation" ?
-p16 (39) and (40) : beta and beta^2 have the same units ?
-p19 l6 "disentrainment rate, consistent with the deposition rate." I do not understand this.
-p20 l25-30 This paragraph is very confusing for me. Could you reformulate it in simpler way ?
-p28 l17 **m** g mu cos theta
-p30 l 24 : What is thus the importance of gamma in a model then ?
-p32-l18 : Why is it problematic ?
-p37 l5-10 : This could have been introduced at the beginning!
-p38 l21 : recall what is alpha

2) Analysis :

The second companion paper present results from an experimental study of particle travel distances down a slope, launched by a catapult system. Data is compared to previous experiments and field studies in an exhaustive manner and tested again the theoretical elements provided in the first companion paper (e.g. the expected Pareto distribution of travel distances). Data is well presented and well detailed so that I believe the 2$^{nd}$ study can be published within minor changes.

First, I do not exactly see why high speed imaging is used apart from determining launched velocity. Indeed, all the results shown in Figures present travel distances that can all be determined without video.

Second, I am not sure to understand how the Pareto fits to the experimental distributions are obtained : by fitting the Pareto parameters, or by estimating them independently with high speed imaging (such as the \beta_z collision restitution parameter) ? I believe the theory would prove very robust if all parameters could be estimated independently via imaging (or other technique). This point is not clear enough and I would suggest the authors to clarify this while presenting their experiments.

Third, it is somewhat disappointing not to see any particle trajectory plotted, that would show the 'heating' (acceleration) for steep slopes, or 'cooling' for milder slopes. I believe much information can be extracted from an acceleration / velocity diagram, as was done for bedload transport in the authors' 2012 paper serie.

Other comments:
Fig 9 and 10 (and maybe others) : recall what is \beta_z in the caption so that each figure is understandable by itself.

3) Entropy :

The third paper has been the hardest for me to follow, because it touches concepts from statistical physics, that are less frequent in the earth science research community. For what I understood, the author claim to generalize the maximum entropy principle to several energy-based physical constraints. With this approach, they find similar Pareto distributions as with the varying deposition probability framework developed in companion paper 1. I believe this is an important result that goes far beyond particle motion on hillslopes, so that I am not convinced that associating this study as a companion paper is a judicious choice. In my opinion, proposing this study to a more physically sound readership journal than esurf would  have a greater impact (Physical review ?). However, I rely on the editor's and other reviewers point on view for this.

Other comments:
p1 l15 "… that is heavy-tailed for net cooling and light tailed for net heating" Isn't it the other way around ?!
(3) precise that A can be between -B and infinity ?
(16) What is notation E[] for ? You have already used it for energy…
p19 l 7: What is Occam's razor ?

*4) Philosophy :*

The fourth paper present a general discussion on probabilistic approach to rarefied particle motions. It correctly points the generality of such approach, and shows how continuum equation of motion extend

(within some subtle extra terms) to ensemble average quantities or probability distributions, even when the instantaneous particle flux is strongly intermittent.

While I completely agree with this viewpoint, and I believe the paper has its importance for the community, I am not sure how this relates specifically to the hillslope motions. Indeed, the use of ensemble averaging/probabilistic description to describe rarefied gas, bedload, or avalanches, and the scale dependence of fluctuations, is a much more general discussion that could fit in a standalone study, with dedicated title. Indeed, the 4$^{th}$ papers format dilutes in my sense the distinct messages the authors convey. Nevertheless, if the editors and reviewers think the inclusion of this paper as a companion paper is justified I will not argue against this.

One minor comment is the following. The authors point 2 equivalent probabilistic viewpoints, the Fokker-Planck equation (the linearization of the master equation) and the maximum entropy approach, originating from statistical physics (they discussed in the 1$^{st}$ and 3$^{rd}$ companion paper). In the discussion, I would include a third way, the Poisson representation [1], which has the attracting characteristic of being exactly equivalent to the Master Equation, while leading to continuous, analytically tractable PDEs. This approach, developed by Gardiner, can be used [1,2] to compute the exact particle number pdf and correlations from basic entrainment/disentrainment rules, without requiring a "small" noise or Kramer-Moyal expansion that assume a large number of particles. As pointed by Gardiner, it has the potential to describe "low density-high fluctuations" states of granular gases, for which large deviations play an important role. A mention of such alternative could be relevant.

1 Gardiner, C. W. (1985). *Handbook of stochastic methods* (Vol. 3, pp. 2-20). Berlin: springer.
2 Ancey, C., & Heyman, J. (2014). A microstructural approach to bed load transport: mean behaviour and fluctuations of particle transport rates. *Journal of Fluid Mechanics, 744,* 129-168.
3 Heyman, J., Ma, H. B., Mettra, F., & Ancey, C. (2014). Spatial correlations in bed load transport: Evidence, importance, and modeling. *Journal of Geophysical Research: Earth Surface, 119*(8), 1751-1767.

---

## Short Comment (SC1) · 15 Feb 2021

In part 2 of the Rarefied paper series, Furbish et al. analyze a combination of previously published field and experimental data, as well as new experimental data, to compare with predictions from the rarefied hillslope sediment transport theory independently developed in the first paper. They first summarize the key theoretical components and predictions from the first paper. I found this section very helpful, as it crystallizes key aspects of the first paper. Next, they step through a series of experimental and field studies to examine different aspects of the theory, and find that all data support their general predictions: particle travel distances can be characterized by a general Pareto distribution, where the specific form of the distribution is controlled by the kinetic energy balance of the particles. They include a useful discussion of limitations of their work,

along with suggestions for future studies that can untangle some of the unknown details of particle behavior. The videos from the Vanderbilt experiments are delightful, and really help to visualize a lot of the concepts presented in the paper. I found their careful analysis convincing and mostly well-presented, and I imagine this paper will become canonical among those studying sediment transport not only on steep hillslopes, but in a variety of settings.

I have only minor comments for this paper. While much of the paper is clear and well-organized, a couple elements remain unclear: 1) the role of grain size/angularity and how it relates to theory 2) the difference between presented experiments and field studies, and why they test different aspects of the theory. To be clear, these points are both discussed extensively in the paper, but without organized explanations both toward the beginning of the paper and at the beginning of each new section, they are a bit hard to follow. One unclear point relates to the subtle difference between spherical particles traveling over a rough surface, and angular particles traveling over a smooth surface, with seemingly similar effects. I think this is one of the more interesting points of the paper, but it is currently lost without being set up properly in the introduction.

Line by line comments:

Page 2, Lines 4-5: can you give an example of another system where this work might be relevant?

Page 2, Lines 10-21: This is an excellent summary of the theory presented in the first paper.

Page 2, Lines 24-25: are you not mainly summarizing theory from the first paper?

Page 2, Line 28: change to "new laboratory experiments" to show that they are being reported for the first time in this study

Page 11, Line 23: "Section 3, Laboratory Measurements": Add a little intro here to remind us where we are. "Now we're going to summarize experimental studies and

compare their results to our theory..." Also consider adding a very brief summary of the results for both experimental and field comparisons, as it will help guide the reader through the various points of comparison in the coming sections...

Figure 3 and 4: What do the different symbols in the figure correspond to?

Page 14 line 8: "bumpety bump" I sincerely hope this technical wording remains in the final verison of the paper.

Page 15 line 17: can you more explicitly state how these experiments differ from those of gabet? What aspects of the theory will you be comparing for this set of experiments?

Page 17: "Experiments" Summarize briefly (1 sentence) what aspects of theory you will be comparing. Did you also measure travel distances to be able to make a plot similar to figures 3 and 4? Oh, I see. Why are figures after figure 10 placed at the end of the manuscript? I'm sure this will be fixed in editing but it currently hides some of the most exciting results of the paper.

Vanderbilt Experiments: Though you test different grain sizes, it is unclear what effect this has in your experiments. Is there a plot you can show highlighting the effect? Or a short amount of discussion?

Page 24, section 4.1: Typo. "DiBiase"

Page 26, line 27: did this experiment endanger helpless banana slugs? I sure hope not.

Page 26, line 31: Phew.

Page 30, lines 8-9: Why not plot the data like this to show us?

Page 33, line 39: It's unclear what this means- can you expand on it a little?

---

## Author Comment (AC1) · 26 Mar 2021

The comment was uploaded in the form of a supplement:
https://esurf.copernicus.org/preprints/esurf-2020-99/esurf-2020-99-AC1-supplement.pdf

---

## Referee Report (RR1)

**Review for revised manuscript: Rarefied particle motions on hillslopes: 2. Analysis**

The authors have adequately addressed my minor comments on the original version of this manuscript. The paper clearly and thoroughly presents field and experimental data that support the theory shown in the first paper, with plenty of discussion for how future studies may build on the work. The compiled datasets are plotted together in the remaining two papers; I think this is justifiable, since the first two papers in essence present a specific example of the broader approach and philosophy presented in the last two. Taken as a whole, in my opinion the four part series should absolutely be published together, as each paper builds upon and broadens the scope of the previous. I have no further changes to suggest.